# MixFP4: Enhancing NVFP4 with Adaptive FP4/INT4 Block Representations

Jiaxiang Zou [1]  Yonghao Chen [1]  Ruilong Wu [1]  Xinyu Chen [1]

## Abstract

As large language models continue to scale, fine-grained block-scaled low-precision formats such as NVFP4 are increasingly adopted for their substantial throughput and memory benefits. However, a single FP4 micro-format often mismatches heterogeneous block-level tensor statistics. To address this without changing the standard block-scaled MMA/GEMM execution path, we propose MixFP4, a mixed micro-format extension to NVFP4 that selects between two stored FP4 micro-formats (E2M1 and E1M2) per block. MixFP4 reuses NVFP4's scale hierarchy and encodes the format choice with zero additional metadata by repurposing the sign bit of the FP8 E4M3 block scale. By decoding both micro-formats into a unified internal E2M2 compute representation, MixFP4 avoids datapath duplication. Across representative LLM families, MixFP4 improves FP4 quantization robustness and accuracy over NVFP4 baselines with modest tensor-core overhead (3.1% area, 1.5% power).

## 1. Introduction

Large Language Models (LLMs) are increasingly deployed in latency and cost sensitive settings (Achiam et al., 2023; Grattafiori et al., 2024; Liu et al., 2024a; Yang et al., 2025), where both compute and memory bandwidth limit end-to-end throughput (Agrawal et al., 2024; Kamath et al., 2025; Zhu et al., 2025; Lin et al., 2026; Zhong et al., 2024; Du et al., 2025). Quantization is therefore indispensable for efficient LLM serving and training. Yet pushing quantization to 4-bit precision remains difficult: with only a handful of representable values, error becomes highly sensitive to local tensor statistics, and naive coarse-grained quantization can easily saturate or inject excessive noise (Cook et al., 2025; Chen et al., 2025a; Guo et al., 2022; Hu et al., 2025a).

To reduce such quantization errors under severe dynamic-range stress, recent hardware and software stacks have moved toward low-bit floating-point (FP) formats (e.g., FP8/FP4), and in particular toward block-scaled FP4 formats such as NVFP4 (Abecassis et al., 2025), which attach a shared scale to each small block to expand effective dynamic range while retaining 4-bit payload storage. However, block scaling alone does not remove the fundamental heterogeneity of LLM tensor distributions: within the same activations or weights, a few rare outliers may coexist with large regions of relatively "flat" small-magnitude values, and the prevalence of these regimes can vary widely across modules, layers, and blocks (Zhao et al., 2024; Xiao et al., 2023). These mixed regimes place conflicting demands on a single 4-bit codebook—outlier-heavy blocks benefit from exponent-rich spacing, whereas flat blocks benefit from more uniform, INT-like spacing. Consequently, enforcing a single, globally fixed FP4 codebook can be suboptimal even within a single layer.

This issue becomes more pronounced when inference-time feature mixing is applied. Structured orthogonal transforms (e.g., Hadamard transforms) can reduce outliers by spreading large coordinates across dimensions (Ashkboos et al., 2024; Liu et al., 2024b; Hu et al., 2025b; Sun et al., 2024), but they also reshape local distributions and may change which low-bit format is preferable under the same selection rule. These observations suggest a key takeaway: at 4-bit precision, *no single FP4 micro-format is universally optimal across all blocks and modules*. Instead, format choices should adapt at fine granularity to local statistics while remaining compatible with general matrix–matrix multiplication (GEMM)–centric execution.

In this paper, we revisit the FP4 design space through the lens of block-wise statistics and introduce *MixFP4*, a mixed-format FP4 extension built on top of NVFP4. MixFP4 keeps the block-scaled structure and tensor-core–friendly execution path, but allows each block to choose between an exponent-heavier FP4 codebook (better for dynamic-range–stressed blocks) and an INT-like FP4 codebook with more uniform spacing (often preferable after mixing reduces crest factors). Critically, MixFP4 stores the per-block choice without additional storage overhead by reusing redundant bits already present in the NVFP4 scaling hierarchy.

[1]The Hong Kong University of Science and Technology (Guangzhou), Guangzhou, China. Correspondence to: Xinyu Chen <xinyuchen@hkust-gz.edu.cn>.

*Proceedings of the $43^{rd}$ International Conference on Machine Learning*, Seoul, South Korea. PMLR 306, 2026. Copyright 2026 by the author(s).

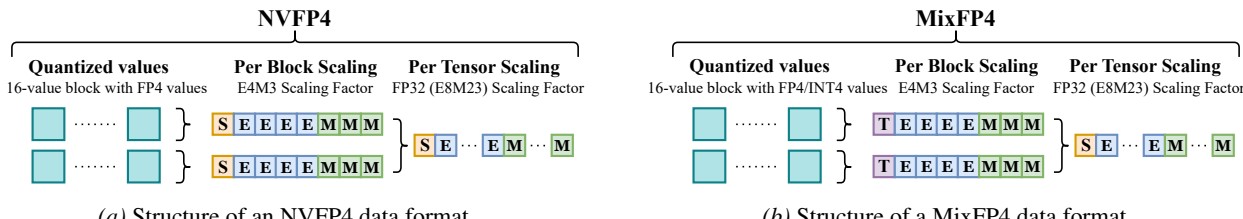

*(a)* Structure of an NVFP4 data format.   *(b)* Structure of a MixFP4 data format.

*Figure 1.* Illustration of NVFP4 and MixFP4 data format structures. MixFP4 reuses the redundant sign bit in the $E4M3$ scale, incurring zero extra storage overhead.

**Contributions.** In this paper, we focus primarily on weight-and-activation quantization to 4-bits per parameter. We propose the following contributions:

- We characterize the heterogeneity of block-wise tensor statistics in modern LLMs and show that orthogonal mixing can reshape these statistics, changing which low-bit codebook is preferred.

- We propose MixFP4, a block-wise mixed micro-format extension to NVFP4 that selects between two FP4 code-books per block, while packing the selection bit into the redundant sign bit of the first-level $E4M3$ block scale to incur zero additional storage overhead.

- We model the hardware overhead required to support MixFP4 on modern GPU Tensor Cores (e.g., Blackwell FP4 Tensor Cores) and show that it is negligible, requiring only lightweight decode and small-format conversion logic while reusing the existing computation datapath.

## 2. Background and Motivation

### 2.1. Introduction to NVFP4 Format

According to the IEEE-754 standard (IEEE, 2019), floating-point (FP) values are represented by a sign bit $S$, an exponent field $E$ with $W_E$ bits, and a mantissa field $M$ with $W_M$ bits. The bias is defined as

$$B = 2^{W_E-1} - 1 \,, \tag{1}$$

and the FP value $x$ can be written as,

$$x = (-1)^S \begin{cases} 2^{E-B}\left(1 + \dfrac{M}{2^{W_M}}\right), & 1 \le E \le 2^{W_E} - 1 \\ 2^{1-B}\left(\dfrac{M}{2^{W_M}}\right), & E = 0 \,. \end{cases} \tag{2}$$

As implied by Equation (2), varying $W_E$ and $W_M$ changes the codebook's level spacing and dynamic range, which in turn affects quantization error for a given local tensor distribution.

NVFP4 is a block-scaled quantization format that stores values in the FP4 E2M1 ($W_E = 2$, $W_M = 1$) micro-format

(intra-block quantized value format). It utilizes a code-book of $\{0, 0.5, 1.0, 1.5, 2.0, 3.0, 4.0, 6.0\}$, supported by a positive FP8 E4M3 per-block scaling factor and an FP32 per-tensor scaling factor. As illustrated in Figure 1a, this two-level scaling scheme operates as follows: (1) a per-tensor FP32 scale maps the entire tensor into the range representable by the block-scaled format (i.e., FP4 with an E4M3 block scale); and (2) a per-block E4M3 scale further maps each block into the specific FP4 E2M1 representable range. For convenience, we define NVINT4 as an ana-log to NVFP4 that replaces the FP4 E2M1 micro-format with symmetric INT4 values, using the integer codebook $\{0, 1, 2, 3, 4, 5, 6, 7\}$ for magnitudes.

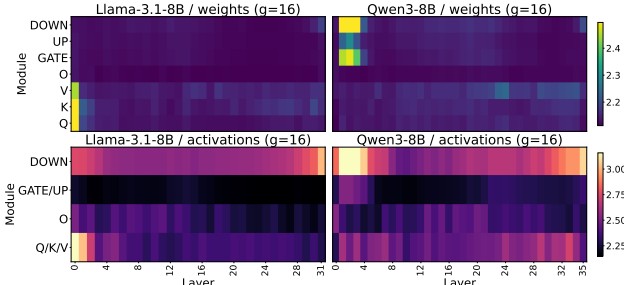

*Figure 2.* Mean within-block crest factor (block size $g=16$) across layers and modules for weights (top) and activation inputs (bottom). Brighter grids indicate modules whose weights or activations have a larger average crest factor, corresponding to sharper (more peaked) within-block value distributions than darker ones. Activations show high spatial variability.

### 2.2. Inter- and Intra-Tensor Statistical Variability

Data distribution diversity is highly heterogeneous across tensor types (weights, activations, and gradients) and across model locations (layers, modules, and even blocks) (Egiazar-ian et al., 2025; Liu et al., 2024b; Ashkboos et al., 2024; Hu et al., 2025b; Sun et al., 2024; Guo et al., 2022; Lee et al., 2025; Lin et al., 2025). To quantify this heterogeneity, we use the *crest factor*, the ratio of the peak absolute value to the root-mean-square (RMS) value, which captures how likely a fixed-step uniform quantizer is to saturate or incur large error. For a block size of 16 with FP8 E4M3 block scale, when the crest factor is below 2.224, NVINT4 yields

higher predicted QSNR (i.e., lower quantization error) than NVFP4; the proof is provided in Appendix A.

Figure 2 illustrates the inter-tensor data diversity, highlighting a distinct contrast: while weight tensors exhibit relatively stable crest factors, activation tensors demonstrate significant spatial variability across both layers and modules. This pronounced spatial heterogeneity indicates that a single, globally fixed low-bit format is unlikely to remain optimal throughout the model.

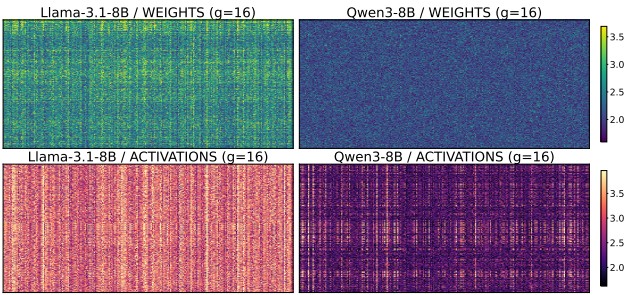

*Figure 3.* Even within a single tensor (Q projection, layer 0; $g$=16), blocks exhibit large variations in crest factor for both weights (top) and activation inputs (bottom) in Llama-3.1-8B and Qwen3-8B, highlighting strong intra-tensor dynamic-range heterogeneity and motivating block-wise format adaptivity. Color indicates within-block crest factor (darker: flatter blocks; brighter: more outlier-prone blocks).

Crucially, this variability extends beyond the inter-layer level down to the intra-tensor domain. As illustrated in Figure 3, even within a single tensor (e.g., Q projection), there is significant block-wise heterogeneity. The heatmaps reveal that different blocks exhibit markedly different crest factors, ranging from "flat" regions to "outlier-prone" hotspots, indicating that a uniform quantization format is insufficient to capture these local dynamic range fluctuations.

**Design Implication.** Low-bit format choices should adapt at fine granularity to local tensor statistics, rather than relying on a single globally fixed format.

### 2.3. One FP4 Micro-format is Not Enough

This sensitivity is especially pronounced at 4-bit precision: because only a small number of discrete levels must represent widely varying local statistics, FP4 naturally forms a family of codebooks in which different $E$ and $M$ splits are better suited to different local tensor distributions. While NVFP4 adopts `E2M1` as its default micro-format, alternative FP4 micro-formats may better fit different local tensor statistics; for example, `E1M2` yields a uniform, INT-like level spacing, while `E3M0` emphasizes power-of-two levels. These options imply a simple but important conclusion: *no single FP4 micro-format is universally optimal across all tensors and all modules.* When tensor statistics vary, a format-aware strategy that adapts to local statistics is a

principled way to reduce quantization error.

Compared with FP4 variants, integer formats can be highly competitive in regimes where the distribution is more uniform. Furthermore, rotation-based Hadamard transforms can reduce crest factors (Ashkboos et al., 2024; Egiazarian et al., 2025; Chen et al., 2025a), uniform INT quantization may become preferable because its level spacing aligns better with near-uniform local statistics. Conversely, in blocks that remain dynamic-range–stressed, exponent-heavy FP variants can retain an accuracy advantage by allocating representable levels more effectively across magnitudes. Taken together, these regimes suggest that the right goal is not "always FP" or "always INT", but rather *fine-grained mixed micro-format* decisions that track local statistics.

**Design Implication.** Block-wise mixed micro-format selection should choose between FP-like and INT-like codebooks to match local statistics.

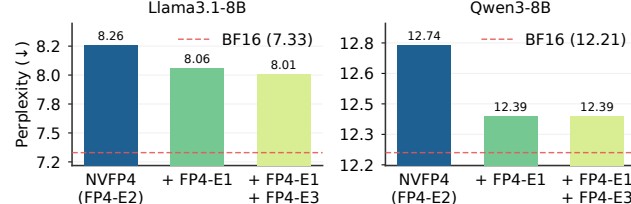

*Figure 4.* Wikitext perplexity for Llama-3.1-8B and Qwen3-8B with NVFP4 quantization. + FP4-E1(refers to `E1M2`) and + FP4-E3(refers to `E3M0`) indicate adding new data formats in NVFP4 quantization.

### 2.4. Format Proliferation Has Diminishing Returns

Although fine-grained mixed micro-format choices can reduce quantization error, expanding the supported format set quickly yields diminishing accuracy returns while incurring growing system-level costs. On the accuracy side, once a small set of representative formats covers the dominant dynamic-range regimes, adding additional, closely related variants typically yields only marginal gains. As shown in Figure 4, we conduct an ablation over three FP4 candidate sets under the block-wise MSE-based selection criterion adopted in prior work (Cook et al., 2025). The results show that extending NVFP4 with `E1M2` yields a substantial accuracy improvement, whereas further adding `E3M0` provides only marginal additional gains.

To further evaluate the benefit of integrating different data formats and determine which formats are better, we select two candidate sets: {FP4 `E2M1`(6), FP4 `E2M1`(4)} (Cook et al., 2025) and {FP4 `E2M1`(6), FP4 `E1M2`, FP4 `E3M0`}. FP4 `E2M1`(4) means use 4 in FP4 `E2M1` as the max quantized value instead of 6. As shown in Figure 5, enlarging the candidate set leads to highly skewed block-wise selections: a small subset of formats dominates (notably `E2M1`(6) and `E1M2`), while others are rarely chosen, and this skew be-

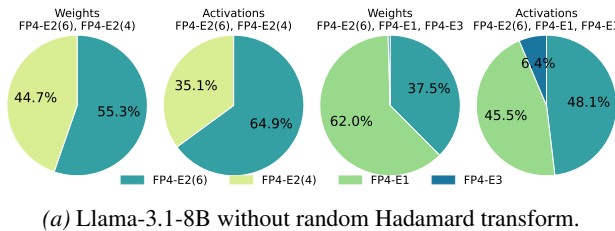

*(a)* Llama-3.1-8B without random Hadamard transform.

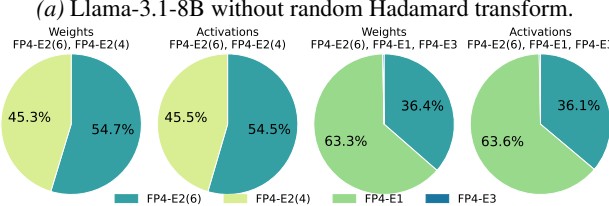

*(b)* Llama-3.1-8B with random Hadamard transform.

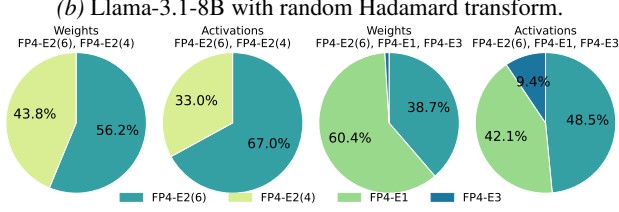

*(c)* Qwen3-8B without random Hadamard transform.

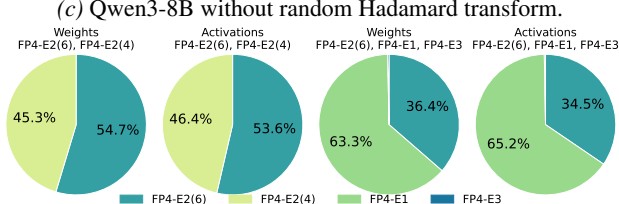

*(d)* Qwen3-8B with random Hadamard transform.

*Figure 5.* Format selection in Llama-3.1-8B and Qwen3-8B, comparing runs without and with random Hadamard transform.

comes even stronger under random Hadamard transforms. This observation motivates a practical design choice: retaining only a small, representative set of formats (E2M1(6) and E1M2) captures most of the accuracy gains, while avoiding the software and hardware overhead of supporting rarely selected variants. Appendix E further visualizes per-tensor block-wise selections and relates them to the crest-factor heterogeneity shown in Figure 3.

**Design Implication.** Retaining only a small, representative set of candidate formats can capture most accuracy gains without format proliferation.

### 2.5. Design Goal: Fine-grained Adaptivity with GEMM-friendly Execution

To translate numerical improvements into end-to-end speedups, the chosen formats must map cleanly onto the GEMM-centric execution of modern accelerators. Recent accelerator designs have similarly shown that numerical formats and arithmetic units should be co-designed for efficient low-precision GEMM execution (Hu et al., 2025a;

Chen et al., 2025b; Zou et al., 2025; Chen et al., 2025c).

Our goal is to improve the low-bit numerical types used by the arithmetic units directly, preserving the standard MMA/GEMM computation pattern with only lightweight front-end decode and block-scaling logic.

Taken together, these observations motivate three design requirements: (i) *fine-grained format adaptivity* to match local tensor statistics, (ii) *a small number of candidate formats* to avoid format proliferation, and (iii) *hardware-friendly execution* that reuses existing GEMM pipelines. Block-scaled FP4 already operates in this regime, and our MixFP4 extension built on top of NVFP4 further adds block-wise adaptivity while retaining GEMM-friendly execution.

## 3. Design

### 3.1. FP4 Encoding and INT4 Compatibility

Driven by the observation in Section 2.4 that block-wise selection is highly skewed towards a few dominant types, our MixFP4 specifically supports two 4-bit payload micro-formats: E2M1 and E1M2. Table 1 summarizes their $S.E.M$ bit layouts and numerical ranges (including zeros and subnormals). We focus on this pair because E2M1 is the NVFP4 payload, while E1M2 provides an INT-like, uniform level spacing that enables INT4-compatible behavior under the same block-scaling hierarchy.

*Table 1.* Details of FP4 Binary Formats

|  | E2M1 | E1M2 |
|---|---|---|
| Exponent bias | 1 | 0 |
| Zeros | $S.00.0$ | $S.0.00$ |
| Max normal | $S.11.1 = 1.5 * 2^2 = 6$ | $S.1.11 = 1.75 * 2 = 3.5$ |
| Min normal | $S.01.0 = 1$ | $S.1.00 = 2$ |
| Max subnormal | $S.00.1 = 0.5$ | $S.0.11 = 0.75 * 2$ |
| Min subnormal | $S.00.1 = 0.5$ | $S.0.01 = 0.25 * 2$ |

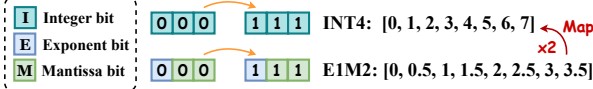

*Figure 6.* The uniform-step E1M2 codebook can represent symmetric INT4 values via scaling.

A key observation is that the stored E1M2 magnitudes form a uniform codebook $\{0, 0.5, 1.0, \ldots, 3.5\}$, whereas symmetric INT4 uses integer levels $\{0, 1, \ldots, 7\}$ in the unsigned domain. MixFP4 bridges these lattices with a fixed $\times 2$ remapping of the E1M2 magnitude, yielding an exact INT4-like lattice $0, 1, \ldots, 7$ (Figure 6). Importantly, this remapping is a constant factor absorbed in the value interpretation at decode time, so the stored FP4 payload and the per-block E4M3 scaling convention remain unchanged, while E1M2 effectively behaves as a true uniform-step INT4 quantizer when block statistics favor INT-like spacing.

**Algorithm 1** MixFP4 data type quantization algorithm.

1: **Input:** Tensor, $X$.
2: **Output:** $\bar{X}$, $s_{32}$, $s_8$.
3: **def** MixFP4($X$)
4: $s_{32} = \frac{max(|X|)}{2688}$ {6 * 448 = 7 * 384 = 2688 (Note: 448 and 384 are E4M3 range)}
5: $X_{\text{FP8}} = \frac{X}{s_{32}}$
6: ―――――――――――――――――
7: $s_{8,\text{E2M1}} = \text{FP8\_E4M3}\left(\frac{\text{block-max}(|X_{\text{FP8}}|)}{6}\right)$ {6 is the largest MixFP4 E2M1 value}
8: $X_{\text{E2M1}}^q = \text{FP4\_E2M1}\left(\frac{X_{\text{FP8}}}{s_{8,\text{E2M1}}}\right)$
9: $D_{\text{E2M1}} = X_{\text{E2M1}}^q \cdot s_{8,\text{E2M1}}$
10: $Err_{\text{E2M1}} = \text{MSE}(D_{\text{E2M1}}, X_{\text{FP8}})$
11: ―――――――――――――――――
12: $s_{8,\text{E1M2}} = \text{FP8\_E4M3}\left(\frac{\text{block-max}(|X_{\text{FP8}}|)}{7}\right)$ {7 is the largest MixFP4 E1M2 value}
13: $X_{\text{E1M2}}^q = \text{FP4\_E1M2}\left(\frac{X_{\text{FP8}}}{s_{8,\text{E1M2}}}\right)$
14: $D_{\text{E1M2}} = X_{\text{E1M2}}^q \cdot s_{8,\text{E1M2}}$
15: $Err_{\text{E1M2}} = \text{MSE}(D_{\text{E1M2}}, X_{\text{FP8}})$
16: ―――――――――――――――――
17: **if** $Err_{\text{E2M1}} < Err_{\text{E1M2}}$ **then**
18: $\quad s_8 = s_{8,\text{E2M1}}$
19: $\quad \bar{X} = X_{\text{E2M1}}^q$
20: **else**
21: $\quad s_8 = s_{8,\text{E1M2}}$ {Set T = 1 at the same time}
22: $\quad \bar{X} = X_{\text{E1M2}}^q$
23: **end if**

### 3.2. MixFP4: Dual Micro-formats in One Block-scaled Format

MixFP4 extends NVFP4 with fine-grained, block-wise mixed-format capability: for each block, it selects between two FP4 micro-formats (E2M1 and E1M2) to better match local tensor statistics, while preserving the standard block-scaled MMA/GEMM execution path. The key idea is to reuse the existing NVFP4 scale hierarchy and encode the per-block format choice with zero additional metadata.

Figure 1 illustrates the underlying block-scaled structure. NVFP4 uses E2M1 to encode the 4-bit payload, E4M3 to encode the per-block (first-level) scale, and FP32 to encode the per-tensor (second-level) scale. MixFP4 retains the same scaling hierarchy and granularity, but allows the payload to be either E2M1 or E1M2. To enable block-wise mixed-format selection with zero additional storage overhead, we store the format type T as a single, block-shared bit by repurposing the underutilized sign bit of the E4M3 first-level scale; since only one such bit is available, MixFP4 supports exactly two formats.

As shown in Algorithm 1, MixFP4 quantizes each tensor block by evaluating the mean squared error (MSE)

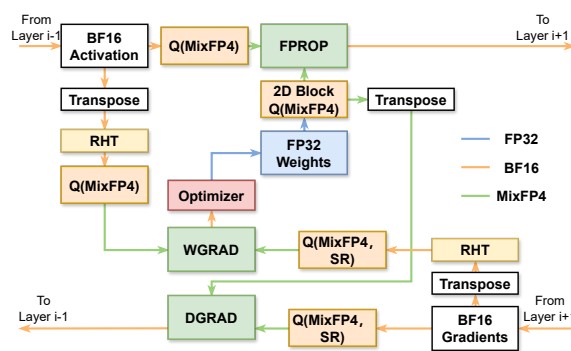

*Figure 7.* Computational flow of a MixFP4 quantized linear layer training. The core matrix multiplications (**FPROP**, **DGRAD**, **WGRAD**) are executed in simulated MixFP4 (green paths), while master weights are maintained in FP32 (blue), and activations/gradients are kept in BF16 (orange). **Q(MixFP4)** represents the proposed quantization mapping blocks to E2M1 or E1M2. Additionally, Stochastic Rounding (**SR**) and random Hadamard transform (**RHT**) are applied to stabilize the backward pass.

induced by each candidate micro-format under the same block-scaling procedure, and selecting the one that yields lower MSE. Concretely, for each block we compute the candidate scale for E2M1 and E1M2, quantize/dequantize the block under each choice, and then compare their MSE. We adopt MSE as the selection criterion since it has been empirically shown to be more reliable than alternative metrics for block-wise format selection (Cook et al., 2025).

Figure 7 shows how MixFP4 is integrated into a quantized linear layer during training. Following recent FP4 training recipes (Abecassis et al., 2025), we apply a random Hadamard transform to the inputs of the weight-gradient computation and perform 2D block quantization on weight matrices. We quantize weights, activations, and gradients to simulate MixFP4 at the GEMM boundaries while keeping high-precision states (e.g., master weights and optimizer states) unchanged. The pre-training results are reported in Section 4.2.

*Table 2.* Comparison of multiplier configurations for multi-precision tensor cores with a target BF16/FP8/FP4 throughput ratio of 4:8:16.

| Design | Multiplier Composition[†] |
|---|---|
| Baseline Unit | $4 \times \text{E8M10} + 4 \times \text{E5M3} + 8 \times \text{E2M1}$ |
| Our Unit | $4 \times \text{E8M10} + 4 \times \text{E5M3} + 8 \times \text{E2M2}$ |

[†] To reflect hardware reuse in tensor cores across varying E/M configurations, we model the FP8 lane with an E5M3 multiplier (covering E4M3/E5M2) and the BF16/FP16 lane with an E8M10 multiplier (covering FP16 E5M10 and BF16 E8M7).

### 3.3. MixFP4 Enabled Tensor Core Design

We target a tensor-core slice that already supports block-scaled FP4 MMA with NVFP4, where values are encoded as

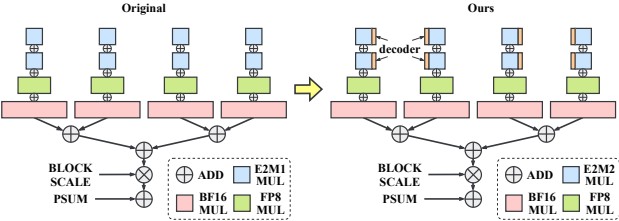

*Figure 8.* MixFP4 supported tensor core.

`E2M1` and each block is scaled by FP8 (`E4M3`). We extend the slice to additionally support an alternative FP4 micro-format (`E1M2`) subject to three invariants—FP16/FP32 accumulation, block scaling granularity, and scale storage footprint—remaining unchanged, making the extension compatibility-preserving rather than a datapath redesign. The motivation is twofold. First, different FP4 codebooks (`E2M1` vs. `E1M2`) can yield materially different error profiles for the same local statistics. Second, `E1M2` provides INT-like uniform level spacing under block scaling, complementing `E2M1` when statistics become more uniform (e.g., after orthogonal mixing).

To support mixed `E2M1`/`E1M2` operands without introducing format-dependent compute overhead (e.g., separate arithmetic datapaths), we normalize both micro-formats into a unified `E2M2` internal representation at decode time. We repurpose the block-scale sign bit as a block-shared type signal, decode `E2M1` by zero-padding its missing mantissa bit, and decode `E1M2` through a small remapping lookup, such that both paths produce the same `E2M2` value (Figure 9, Appendix B.5.1). This unification provides a single `E2M2` data format for downstream computation, avoiding the need for separate compute datapaths for different formats.

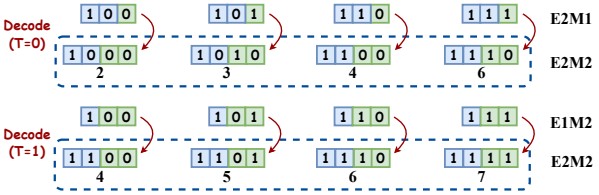

*Figure 9.* Decode `E2M1` or `E1M2` to `E2M2` by `T`.

We model the tensor core as a 3D array with normalized throughput BF16:FP8(FP6):FP4 = 4:8:16, corresponding to $4 \times 4 \times 4$, $4 \times 4 \times 8$, and $4 \times 4 \times 16$ MMA per cycle, respectively (Table 2). To focus on the reduction dimension, we analyze a single output lane by setting $M = N = 1$ (a K-element vector MAC), following prior multi-precision reuse modeling (Chen et al., 2025a). As shown in Figure 8, the baseline slice comprises four BF16/FP16 MAC units, four FP8 MAC units, and eight FP4 (`E2M1`) MAC units; in our MixFP4-enabled slice, we replace the eight `E2M1` MAC units with eight `E2M2` MAC units, each pre-

ceded by lightweight per-lane decode logic for dual-mode (`E2M1`/`E1M2`) support and 4 accumulators between `E2M2` MAC units are extended from `E4M3` to `E4M5`, while reusing the rest of the datapath unchanged.

The incremental hardware cost of supporting `E1M2` is confined to the FP4 decode and the block-scaling control and metadata path. Specifically, the decoding introduces only small combinational logic (e.g., a few multiplexers and simple gates) to remap FP4 bit-fields under a block-shared type signal, without requiring changes to the FP16/FP32 accumulation datapath. We quantify this overhead at the gate level and show it to be negligible relative to the baseline tensor-core slice; detailed results are provided in Appendix B.4.2.

## 4. Evaluation

In this section, we evaluate how MixFP4 affects the performance of models quantized to fine-grained 4-bit formats during post-training and pre-training quantization. Following prior quantization work, we quantize GEMM-heavy linear projections (e.g., Q/K/V/O, MLP up/gate/down) except embedding, LM head, Attention and nonlinear modules. We observe improved stability in our Qwen3-style 114M pre-training setting, and improving downstream performance of already-trained Llama (Grattafiori et al., 2024), Qwen (Yang et al., 2025) and Mamba (Gu & Dao, 2024) models on a wide variety of tasks. We evaluate wikitext (Merity et al., 2016) word perplexity and downstream tasks accuracy on $lm\_eval\_harness$ (Gao et al., 2024). The downstream tasks include MMLU (Hendrycks et al., 2020), ARC-Easy and ARC-Challenge (Clark et al., 2018), HellaSwag (Zellers et al., 2019) and Piqa (Bisk et al., 2020). All software experiments are run on a single H100 GPU with 80GB memory.

For hardware performance, we implement our MixFP4-enabled tensor core in Verilog and evaluate its area and energy overhead in Synopsys Design Compiler targeting TSMC 28nm at 1GHz (TT corner at 0.9V and 25°C, CCS timing model).

### 4.1. Post-training Quantization

We evaluate MixFP4's ability to improve NVFP4 post-training quantization (PTQ) accuracy. We compare with the SOTA baseline 4/6 (Cook et al., 2025), which also tries to improve the accuracy of NVFP4. While MixFP4 can be used on its own, it is a general method that modifies the underlying NVFP4 quantization algorithm, allowing it to be easily combined with existing PTQ methods such as SmoothQuant (Xiao et al., 2023), GPTQ (Frantar et al., 2022), and SpinQuant (Liu et al., 2024b).

Table 3 reports the post-training quantization (PTQ) results on WikiText in terms of perplexity (lower is better) with round-to-nearest (RTN) quantization method. We compare

*Table 3.* The wikitext word perplexity result of MixFP4 and baseline methods with round-to-nearest quantization.

| Model | Llama-3.2-1B | +RHT | Llama-3.1-8B | +RHT | Qwen3-4B | +RHT | Qwen3-8B | +RHT | Qwen3-14B | +RHT | Mamba-1.4B | Mamba-2.8B |
|---|---|---|---|---|---|---|---|---|---|---|---|---|
| BF16 | 11.57 | 11.56 | 7.33 | 7.33 | 16.45 | 16.46 | 12.21 | 12.22 | 10.78 | 10.79 | 13.58 | 11.79 |
| NVFP4 | 13.90 | 15.09 | 8.26 | 8.55 | **17.34** | 19.22 | 12.74 | 13.57 | 11.31 | 11.62 | 14.56 | 12.44 |
| NVINT4 | 14.79 | 13.99 | 8.37 | 8.23 | 18.88 | **18.11** | 12.73 | 12.94 | 11.31 | **11.21** | 14.88 | 12.55 |
| 4/6 | 13.60 | 14.19 | 8.15 | 8.29 | 17.61 | 18.32 | 12.56 | 13.48 | 11.21 | 11.36 | 14.45 | 12.34 |
| MixFP4 | **13.48** | **13.53** | **8.06** | **8.10** | 17.42 | 18.60 | **12.39** | **12.77** | **11.03** | 11.44 | **14.38** | **12.27** |

MixFP4 against BF16 and several representative 4-bit base-lines, including NVFP4, NVINT4, and Four-over-Six (4/6). For Transformer-based models, we additionally report an ablation with random Hadamard transform (+RHT). Overall, MixFP4 achieves the lowest (or near-lowest) perplexity across most models, and is consistently better than the original NVFP4, NVINT4 and 4/6 under the same setting. Notably, +RHT can substantially change the relative ranking of quantizers: while Hadamard mixing often benefits codebooks that are sensitive to outliers, MixFP4 remains competitive and frequently retains the best accuracy, indicating that its mixed codebook design is robust to distribution shifts induced by orthogonal transforms. In a few settings (e.g., Qwen3-4B), NVFP4 or NVINT4 attains the best perplexity; nevertheless, the proposed MixFP4-enabled tensor core remains compatible with these original formats, supporting NVFP4 and NVINT4 in the same execution pipeline without additional format-specific hardware cost.

Table 4 presents the PTQ results on wikitext word perplexity and downstream task accuracy for Llama-3.2-1B, Llama-3.1-8B, Qwen3-4B, and Qwen3-8B, combined with SmoothQuant, GPTQ, and SpinQuant. Detailed experimental settings are shown in Appendix C. The results demonstrate that MixFP4 consistently yields superior performance, achieving the lowest perplexity and the highest average scores in the majority of configurations. [1]

These results also highlight that MixFP4 is complementary to existing calibration and rotation-based PTQ pipelines rather than tied to a single front-end algorithm. SmoothQuant, GPTQ, and SpinQuant change the tensor statistics in different ways, yet MixFP4 usually retains the best or near-best perplexity and average downstream score after being inserted as the underlying 4-bit block format. The few exceptions do not require a separate execution path, since the proposed tensor core remains compatible with the original NVFP4/NVINT4 formats. This compatibility is important for deployment: a system can select the numerically preferred format per model or layer while keeping the same mixed-format execution datapath.

---

[1]We do not show the results of Qwen3-4B with SpinQuant because the model has an intermediate size of 9728 = 38 * 256, which is not supported by SpinQuant (SpinQuant only supports $n$ to be $2^k$ or $K \times 2^k$, but there is no $K = 38$ in its source code).

*Table 4.* PTQ performance of MixFP4 on Llama3 and Qwen3 models combined with various quantization methods.

| | | | | | Llama-3.2-1B | | | |
|---|---|---|---|---|---|---|---|---|
| Method | Task | wiki | MMLU | Arc-C | Arc-E | Hella. | Piqa | Avg. |
| | BF16 | 11.57 | 36.71 | 36.43 | 60.31 | 63.70 | 74.65 | 54.36 |
| Smooth-Quant | NVFP4 | 13.85 | 29.59 | 34.22 | 55.93 | 59.78 | 72.03 | 50.31 |
| | 4/6 | 13.50 | 28.41 | 34.04 | 55.81 | 59.72 | 70.89 | 49.78 |
| | MixFP4 | **13.38** | 31.70 | 33.45 | 55.26 | 60.37 | 72.42 | **50.64** |
| GPTQ | NVFP4 | 13.24 | 30.69 | 33.96 | 56.36 | 59.85 | 72.96 | 50.76 |
| | 4/6 | 13.04 | 31.84 | 34.81 | 57.70 | 60.11 | 71.33 | **51.16** |
| | MixFP4 | **12.91** | 30.96 | 34.13 | 57.01 | 59.75 | 72.87 | 50.97 |
| Spin-Quant | NVFP4 | 14.44 | 29.35 | 33.11 | 53.96 | 58.93 | 68.44 | 48.76 |
| | 4/6 | 13.69 | 27.05 | 33.96 | 55.18 | 59.18 | 69.53 | 48.98 |
| | MixFP4 | **13.38** | 27.92 | 32.59 | 57.07 | 60.24 | 69.04 | **49.37** |
| | | | | | Llama-3.1-8B | | | |
| Method | Task | wiki | MMLU | Arc-C | Arc-E | Hella. | Piqa | Avg. |
| | BF16 | 7.33 | 63.38 | 53.24 | 81.10 | 78.92 | 81.07 | 71.54 |
| Smooth-Quant | NVFP4 | 8.22 | 60.53 | 50.17 | 76.68 | 77.36 | 79.71 | 68.89 |
| | 4/6 | 8.10 | 59.67 | 51.54 | 78.16 | 77.64 | 80.30 | **69.46** |
| | MixFP4 | **8.03** | 60.79 | 50.26 | 78.11 | 77.96 | 79.65 | 69.35 |
| GPTQ | NVFP4 | 8.24 | 59.93 | 49.83 | 75.00 | 77.46 | 79.82 | 68.41 |
| | 4/6 | 8.16 | 60.60 | 51.19 | 79.04 | 77.86 | 80.09 | **69.76** |
| | MixFP4 | **8.05** | 60.67 | 50.60 | 76.73 | 77.96 | 79.92 | 69.18 |
| Spin-Quant | NVFP4 | 8.66 | 55.85 | 50.34 | 77.61 | 76.33 | 78.35 | 67.70 |
| | 4/6 | 8.43 | 58.59 | 50.34 | 76.73 | 76.76 | 78.02 | 68.09 |
| | MixFP4 | **8.26** | 59.36 | 52.73 | 77.74 | 77.38 | 78.84 | **69.21** |
| | | | | | Qwen3-4B | | | |
| Method | Task | wiki | MMLU | Arc-C | Arc-E | Hella. | Piqa | Avg. |
| | BF16 | 16.45 | 68.35 | 53.92 | 78.41 | 68.44 | 75.19 | 68.86 |
| Smooth-Quant | NVFP4 | 17.90 | 65.65 | 49.40 | 72.31 | 65.92 | 73.78 | 65.41 |
| | 4/6 | 17.66 | 65.99 | 50.00 | 73.36 | 66.58 | 74.10 | 66.01 |
| | MixFP4 | **17.52** | 66.98 | 51.11 | 76.05 | 66.90 | 73.56 | **66.92** |
| GPTQ | NVFP4 | 17.20 | 65.99 | 49.32 | 73.40 | 66.06 | 73.07 | 65.57 |
| | 4/6 | **17.08** | 66.19 | 50.85 | 74.45 | 66.35 | 73.45 | 66.26 |
| | MixFP4 | 17.13 | 66.27 | 50.85 | 75.08 | 66.97 | 74.43 | **66.72** |
| | | | | | Qwen3-8B | | | |
| Method | Task | wiki | MMLU | Arc-C | Arc-E | Hella. | Piqa | Avg. |
| | BF16 | 12.21 | 73.00 | 56.14 | 80.68 | 74.87 | 77.53 | 72.44 |
| Smooth-Quant | NVFP4 | 12.72 | 71.13 | 54.86 | 80.18 | 73.09 | 75.95 | 71.04 |
| | 4/6 | 12.72 | 70.80 | 54.86 | 78.70 | 73.58 | 76.39 | 70.87 |
| | MixFP4 | **12.48** | 71.42 | 55.55 | 79.55 | 73.75 | 76.28 | **71.31** |
| GPTQ | NVFP4 | 12.68 | 71.31 | 54.27 | 80.47 | 72.92 | 76.50 | 71.09 |
| | 4/6 | 12.60 | 71.56 | 56.91 | 79.17 | 73.33 | 76.17 | **71.43** |
| | MixFP4 | **12.44** | 71.04 | 55.63 | 78.75 | 73.46 | 76.12 | 71.00 |
| Spin-Quant | NVFP4 | 13.12 | 67.50 | 50.26 | 73.44 | 70.47 | 74.32 | 67.20 |
| | 4/6 | **12.96** | 69.14 | 51.19 | 72.90 | 71.97 | 72.80 | 67.60 |
| | MixFP4 | 12.98 | 70.45 | 53.33 | 77.99 | 72.66 | 74.48 | **69.78** |

## 4.2. Pre-training Quantization

We train two Qwen3-style decoder-only models from scratch on the FineWeb-Edu dataset (Penedo et al., 2024) using our method and baseline methods. Both models use Qwen3's tokenizer and $QwenForCausalLM$ from Huggingface, with RoPE (Su et al., 2024), SwiGLU (Shazeer, 2020), and Query-Key normalization (Henry et al., 2020). The model in Figure 10 has 114M parameters and is trained for 2.5B tokens; it uses a hidden size of 512, 8 query heads,

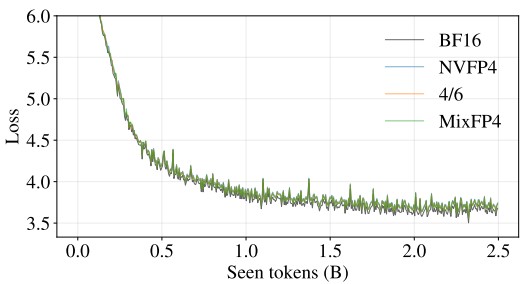

*(a)* Training loss over the full pre-training run.

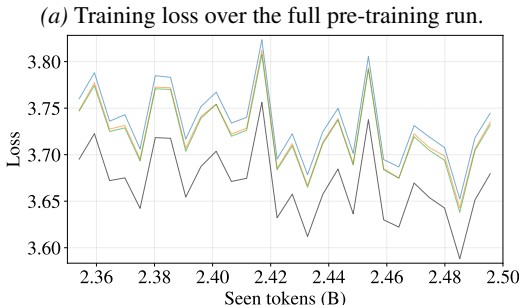

*(b)* Zoom-in of the final pre-training stage (2.35B–2.50B tokens).

*Figure 10.* Pre-training loss of the 114M Qwen3-style model under BF16 and FP4 variants: (a) full run; (b) zoom-in.

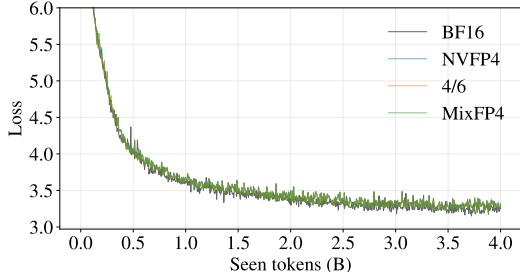

*(a)* Training loss over the full 4B-token run.

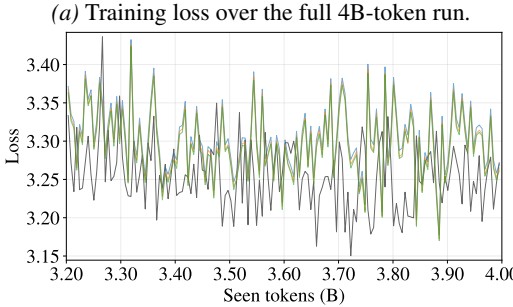

*(b)* Zoom-in of the final pre-training stage (3.20B–4.00B tokens).

*Figure 11.* Pre-training loss of the 476M Qwen3-style model under BF16 and FP4 variants: (a) full run; (b) zoom-in.

4 key-value heads, intermediate size 2048, and 9 layers. To further evaluate scale, Figure 11 uses the larger 476M-parameter setting from our training code, with hidden size 1024, 16 query heads, 4 key-value heads, intermediate size 4096, and 18 layers, and is trained for 4B tokens.

For both settings, we use AdamW (Loshchilov & Hutter, 2017) with $\beta_1 = 0.9$ and $\beta_2 = 0.95$, weight decay 0.1, gradient clipping at 1.0, sequence length 2048, and global batch size 256. The 114M run uses a maximum learning rate of $1 \times 10^{-3}$ and minimum learning rate of $1 \times 10^{-4}$, while the 476M run uses a maximum learning rate of $6 \times 10^{-4}$ with the same 0.1 minimum-learning-rate ratio.

Following from the current state-of-the-art NVFP4 pre-training recipe (Abecassis et al., 2025), we perform stochastic rounding on gradients, apply a random Hadamard transform to both inputs of the weight gradient calculation, and perform 2D block quantization on weight matrices, as outlined in Figure 7. We find stochastic rounding is also useful in MixFP4 enabled pre-training. We report the effects of stochastic rounding in more detail in Appendix D.

Figures 10 and 11 show the pre-training loss trajectories when MixFP4 is applied throughout training. In the 114M setting, the original NVFP4 baseline shows unstable optimization, whereas MixFP4 maintains a smoother loss trajectory and reaches a lower final loss than 4/6. The larger 476M experiment shows the same qualitative behavior: MixFP4 remains consistently below the baselines in the late training stage, suggesting that the gain is not limited to the small

pilot model. These results suggest that block-wise microformat selection can improve the numerical robustness of FP4 pre-training in the evaluated settings.

## 4.3. Block-size Sensitivity

All main software experiments use block size $g = 16$, matching the fine-grained NVFP4 setting targeted by MixFP4. Table 5 ablates the block size on WikiText perplexity for Llama-3.1-8B and Qwen3-8B.

*Table 5.* Block-size sensitivity on WikiText perplexity (lower is better).

| Model | BS | FP4-E2 | +FP4-E1 | +FP4-E3 | +E1+E3 |
|---|---|---|---|---|---|
| Llama-3.1-8B | 8 | 8.04 | 7.83 | 8.00 | 7.79 |
| | 16 | 8.26 | 8.06 | 8.21 | 8.01 |
| | 32 | 8.49 | 8.34 | 8.40 | 8.27 |
| | 64 | 8.73 | 8.69 | 8.63 | 8.54 |
| Qwen3-8B | 8 | 12.48 | 12.39 | 12.46 | 12.40 |
| | 16 | 12.74 | 12.39 | 12.63 | 12.39 |
| | 32 | 12.82 | 12.78 | 12.77 | 12.73 |
| | 64 | 13.00 | 12.96 | 12.85 | 12.79 |

At block size $g = 16$, FP4-E2+FP4-E1 is the best two-format NVFP4 extension, nearly matching the full mixture. This supports our scale-bit reuse design: the extra type bit lets each NVFP4 block choose between FP-like E2M1 and INT-like E1M2 without extra storage. But at larger block sizes, FP4-E1 weakens because each shared scale spans

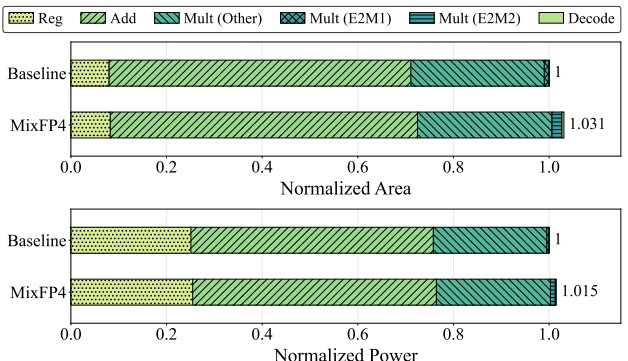

*Figure 12.* Relative area and power overhead of MixFP4 compared to the baseline tensor core.

more heterogeneous values; under this coarser scaling granularity, FP4-E2+FP4-E3 becomes more favorable, as the exponent-heavy E3M0 format better accommodates blocks with wider dynamic range.

### 4.4. Hardware Evaluation

For hardware performance, Figure 12 compares MixFP4 with the baseline tensor-core slice in terms of normalized area and power overhead, focusing on the compute datapath. We partition the datapath into FP adders (Add), FP multipliers (Mult), the decode logic (Decoder) and registers (Reg); the multipliers are further grouped into the baseline FP4 `E2M1` lane (E2M1), the MixFP4 FP4 `E2M2` lane (E2M2), and other multipliers such as FP8 and BF16 (Other). Overall, MixFP4 introduces only a 3.1% area overhead and a 1.5% power overhead relative to the baseline, demonstrating high hardware efficiency despite enabling block-wise mixed-format execution. Since these overheads are measured on the tensor-core compute datapath, non-compute components in a full tensor core and the rest of the GPU (e.g., on-chip storage and control logic) further dilute the chip-level impact. Furthermore, the bar composition indicates that the added cost is concentrated in the existing arithmetic and decode portions, consistent with a localized extension rather than duplicated compute pipelines. Accordingly, although our synthesis targets 28nm at 1GHz, the changes are incremental and dominated by local arithmetic logic (as opposed to large memory structures or global interconnect), so we expect similar relative overhead under technology scaling to advanced GPU processes.

### 5. Discussion

MixFP4 is designed to extend the standard NVFP4 format and hardware co-design by enabling an INT-style uniform-step format at block granularity. Our current prototype is implemented in PyTorch to faithfully simulate quantization accuracy. Native tensor-core kernel latency/throughput for MixFP4 execution depends on hardware support and kernel engineering that are outside the scope of this paper. We will leave it for our future work.

### 6. Conclusion

This paper proposes MixFP4, a tensor core co-designed extension of NVFP4. By reusing redundant bits in the E4M3 scale, MixFP4 introduces zero storage overhead while resolving the mismatch between a single fixed FP4 codebook and heterogeneous block-level statistics. Specifically, it selects E2M1 or E1M2 per block under a unified block-scaling pipeline, leveraging FP-like and INT-like distributions to better match local data distributions. Experiments show that MixFP4 improves robustness and accuracy over representative state-of-the-art NVFP4 baselines in both post-training and pre-training, with only 1.5% power and 3.1% area overhead on the tensor-core compute datapath.

### Impact Statement

This paper presents work whose goal is to advance the field of machine learning. There are many potential societal consequences of our work, none of which we feel must be specifically highlighted here.

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

## A. NVINT4 vs. NVFP4 QSNR Crossover with Exact INT4 Range

This appendix derives the crest-factor crossover $\kappa$ at which NVINT4 and NVFP4 have equal quantization signal-to-noise ratio (QSNR). We refine our proof based on (Chen et al., 2025a). Here $\kappa$ (sometimes denoted $k$ informally) is the *crest factor* of a block:

$$\kappa = \frac{\max_i |X_i|}{\sigma}, \qquad \sigma = \text{RMS}(X). \tag{3}$$

We follow the paper's Gaussian block assumptions $X_i \overset{\text{i.i.d.}}{\sim} \mathcal{N}(0, \sigma^2)$ and the AbsMax scaling convention, but we *do not* use the paper's simplification $Q \approx 2^{b-1}$ for INT4. Instead we keep the exact symmetric INT4 maximum code $Q = 7$.

### A.1. QSNR and relative MSE

QSNR is defined as

$$\text{QSNR} = -10 \log_{10} \left( \frac{\|X - X_q\|_2^2}{\|X\|_2^2} \right). \tag{4}$$

Under the i.i.d. block model, QSNR is well-approximated by a scalar *relative MSE*

$$R = \frac{\text{E}[e^2]}{\text{E}[X^2]} = \frac{\text{E}[e^2]}{\sigma^2}, \qquad e = X - X_q, \qquad \text{QSNR} = -10 \log_{10} R. \tag{5}$$

Therefore, $\text{QSNR}_{\text{NVINT4}}(\kappa) = \text{QSNR}_{\text{NVFP4}}(\kappa)$ is equivalent to

$$R_{\text{NVINT4}}(\kappa) = R_{\text{NVFP4}}(\kappa). \tag{6}$$

### A.2. NVINT4 relative MSE with exact max code $Q = 7$

NVINT4 uses a symmetric signed INT4 codebook $\{-7, -6, \ldots, 6, 7\}$, i.e.

$$Q = 2^{b-1} - 1 = 2^3 - 1 = 7. \tag{7}$$

With AbsMax scaling, the ideal scale maps the block maximum to $\pm Q$:

$$s = \frac{\max_i |X_i|}{Q} = \frac{\kappa \sigma}{Q}. \tag{8}$$

NV formats use a high-precision scale (E4M3 + FP32 second-level), so we take $s' \approx s$ and the quantization step is $\Delta = s'$.

**Approximation A1 (high-resolution uniform error for INT).** For a non-saturating round-to-nearest uniform quantizer, the elementwise error is modeled as $e \sim \text{Uniform}[-\Delta/2, \Delta/2]$, yielding

$$\text{E}[e^2] \approx \frac{\Delta^2}{12}. \tag{9}$$

Substituting $\Delta \approx \kappa \sigma / Q$ gives the baseline INT relative MSE

$$R_{\text{INT}}(\kappa) \approx \frac{1}{12} \left( \frac{\kappa}{Q} \right)^2. \tag{10}$$

**Approximation A2 (NV "one near error-free element" refinement).** Because the per-block scale is high precision for NV formats, the block-maximum element that determines $\max_i |X_i|$ maps (near) exactly to $\pm Q$ and contributes negligible error. For a block of size $g$, this reduces the average error energy by a factor $(g-1)/g$. Thus

$$R_{\text{NVINT4}}(\kappa) \approx \frac{1}{12} \left( \frac{\kappa}{Q} \right)^2 \frac{g-1}{g}. \tag{11}$$

For NV formats, $g = 16$, and for INT4 we enforce $Q = 7$, hence

$$R_{\text{NVINT4}}(\kappa) \approx \frac{\kappa^2}{12 \cdot 7^2} \cdot \frac{15}{16}. \tag{12}$$

## A.3. NVFP4 relative MSE (FP4(E2M1) with subnormals)

NVFP4 uses FP4(E2M1) with subnormals enabled and AbsMax scaling to the largest finite normal magnitude. For FP4(E2M1) (as used in the paper), the constants are:

$$M = 1, \qquad B = 1, \qquad Q_{\max} = 6, \qquad N_{\min} = 2^{1-B} = 1, \qquad S_{\min} = 2^{1-B-M} = 2^{-1} = \tfrac{1}{2}. \tag{13}$$

AbsMax scaling maps $\max_i |X_i|$ to $Q_{\max}$, so with high-precision NV scaling $s' \approx s$:

$$s' \approx s = \frac{\max_i |X_i|}{Q_{\max}} = \frac{\kappa\sigma}{6}. \tag{14}$$

The signal-domain threshold between normal and subnormal regions is

$$T_N := s' N_{\min} = s' \quad \Rightarrow \quad t = \frac{T_N}{\sigma} = \frac{\kappa}{6}. \tag{15}$$

### A.3.1. NORMAL-REGION ERROR TERM

Define the normal-region energy fraction

$$w_{\mathrm{norm}}(\kappa) = \frac{\mathrm{E}[X^2 \, \mathbf{1}\{|X| \geq T_N\}]}{\sigma^2}. \tag{16}$$

**Approximation A1 again (uniform within-cell error in each FP bin).** Within a normal FP bin, the error is modeled as uniform with variance $\Delta(X)^2/12$.

**Approximation A3 (log-phase uniformity).** Let $r := 2^{\{\log_2(|X|/s')\}} \in [1, 2)$ denote the fractional "log-phase". The paper assumes $r$ is approximately uniform on $[1, 2)$, implying $\mathrm{E}[1/r^2] = 1/2$. Under this assumption, the normal-region contribution becomes

$$\frac{\mathrm{E}[e^2 \, \mathbf{1}\{|X| \geq T_N\}]}{\sigma^2} \approx \alpha_M \, w_{\mathrm{norm}}(\kappa), \qquad \alpha_M := \frac{1}{24 \cdot 2^{2M}}. \tag{17}$$

For NVFP4(E2M1), $M = 1$, so

$$\alpha := \alpha_{M=1} = \frac{1}{96}. \tag{18}$$

### A.3.2. SUBNORMAL-REGION ERROR TERM

Let

$$p_{\mathrm{sub}}(\kappa) := \mathrm{P}(|X| < T_N) \tag{19}$$

be the probability of falling into the subnormal region. In the subnormal region, the absolute step is constant:

$$\Delta_{\mathrm{sub}} := s' S_{\min} = s' \, 2^{1-B-M}. \tag{20}$$

Using Approximation A1, $\mathrm{E}[e^2 \mid |X| < T_N] \approx \Delta_{\mathrm{sub}}^2/12$, and since $s' = \kappa\sigma/Q_{\max}$, the subnormal contribution to relative MSE is

$$\frac{\mathrm{E}[e^2 \, \mathbf{1}\{|X| < T_N\}]}{\sigma^2} \approx \beta \, \kappa^2 \, p_{\mathrm{sub}}(\kappa), \qquad \beta := \frac{2^{2(1-B-M)}}{12 \, Q_{\max}^2}. \tag{21}$$

For NVFP4(E2M1), $B = 1$, $M = 1$, $Q_{\max} = 6$, hence

$$\beta = \frac{2^{-2}}{12 \cdot 6^2} = \frac{1}{1728}. \tag{22}$$

### A.3.3. NV REFINEMENT (EXCLUDING THE BLOCK MAXIMUM FROM THE NORMAL BUDGET)

**Approximation A2 again (max element has negligible error).** With high-precision scaling, the block maximum maps (near) exactly to $Q_{\max}$, so the paper refines the normal-region budget by subtracting the energy fraction of the maximum element:

$$\eta := \frac{\max_i |X_i|^2}{g\sigma^2} = \frac{\kappa^2}{g}. \tag{23}$$

Thus the refined NVFP4 relative MSE model is

$$R_{\mathrm{NVFP4}}(\kappa) \approx \alpha \left( w_{\mathrm{norm}}(\kappa) - \frac{\kappa^2}{g} \right) + \beta \, \kappa^2 \, p_{\mathrm{sub}}(\kappa). \tag{24}$$

**A.4. Closed forms for $w_{\mathrm{norm}}(\kappa)$ and $p_{\mathrm{sub}}(\kappa)$ under Gaussian blocks**

Let $Z := X/\sigma \sim \mathcal{N}(0,1)$. Define the standard normal pdf/cdf:

$$\varphi(z) := \frac{1}{\sqrt{2\pi}}e^{-z^2/2}, \qquad \Phi(z) := \int_{-\infty}^{z} \varphi(u)\,du. \tag{25}$$

From (15), $t = \kappa/6$. Then

$$p_{\mathrm{sub}}(\kappa) = \mathrm{P}(|Z| < t) = 2\Phi(t) - 1. \tag{26}$$

Also,

$$w_{\mathrm{norm}}(\kappa) = \mathrm{E}\big[Z^2\,\mathbf{1}\{|Z| \geq t\}\big] = 2\int_{t}^{\infty} z^2\varphi(z)\,dz. \tag{27}$$

We evaluate the integral by parts using $\varphi'(z) = -z\varphi(z)$:

$$\int_{t}^{\infty} z^2\varphi(z)\,dz = [-z\varphi(z)]_{t}^{\infty} + \int_{t}^{\infty} \varphi(z)\,dz$$
$$= t\varphi(t) + \big(1 - \Phi(t)\big). \tag{28}$$

Therefore

$$w_{\mathrm{norm}}(\kappa) = 2\big(t\varphi(t) + 1 - \Phi(t)\big), \qquad t = \kappa/6. \tag{29}$$

**A.5. Crossover equation and numerical solution (enforcing $Q = 7$)**

Equating (12) and (24), with $g = 16$, $\alpha = 1/96$, $\beta = 1/1728$, and $t = \kappa/6$, yields a scalar equation in $\kappa$:

$$\frac{1}{96}\left(2\big(t\varphi(t) + 1 - \Phi(t)\big) - \frac{\kappa^2}{16}\right) + \frac{1}{1728}\kappa^2\,(2\Phi(t) - 1) = \frac{\kappa^2}{12 \cdot 7^2}\cdot\frac{15}{16}, \qquad t = \kappa/6. \tag{30}$$

Because $\Phi(\cdot)$ is involved, (30) has no elementary closed-form solution and is solved numerically (e.g., bisection/Newton).

Solving (30) gives

$$\kappa^* = 2.224277301764024. \tag{31}$$

Which means when $\kappa < \kappa^*$, NVINT4 has better QSNR than NVFP4, and when $\kappa > \kappa^*$, NVFP4 has better QSNR than NVINT4.

At $\kappa = \kappa^*$, the matched relative MSE and QSNR are

$$R^* = R_{\mathrm{NVINT4}}(\kappa^*) = R_{\mathrm{NVFP4}}(\kappa^*) = 0.007888089150418761 \tag{32}$$

$$\mathrm{QSNR}^* = -10\log_{10}(R^*) = 21.03028189684982 \text{ dB} \tag{33}$$

# B. Hardware Overhead Analysis

### B.1. Background: Block-Scaled FP4 as a Factored Computation

A block-scaled FP4 operand $\mathbf{A}$ is represented as a packed FP4 tensor $\mathbf{A}_q$ and a scale tensor $\mathbf{s}_A$ such that

$$\mathbf{A} \approx \mathbf{A}_q \odot \mathbf{s}_A, \tag{34}$$

where $\mathbf{s}_A$ is constant over each block of $V$ values along the reduction dimension ($V = 16$ in this work). For a dot product of length $V$ in one block,

$$\sum_{i=0}^{V-1} (a_i s_A)(b_i s_B) = (s_A s_B)\sum_{i=0}^{V-1} a_i b_i. \tag{35}$$

Equation (35) motivates a microarchitecture that computes an unscaled partial dot $\sum a_i b_i$ first, and applies a single scale-product multiply per block. This reduces scaling-related arithmetic by a factor of $V$ relative to per-element dequantization.

## B.2. Unified Dual-Mode FP4: NVFP4-E2 and NVFP4-E1

We define two block-scaled FP4 modes that share block size and scale type:

- **NVFP4-E2**: block size=16, s=E4M3, FP4 type=E2M1.

- **NVFP4-E1**: block size=16, s=E4M3, FP4 type=E1M2.

**FP4 bit layout.** Each FP4 datum uses one sign bit and a 3-bit payload:

$$x_4 = [s \mid p_2 p_1 p_0].$$

The payload is interpreted differently depending on the mode:

$$\texttt{E2M1}: \quad e = [p_2 p_1], \quad m = [p_0] \tag{36}$$

$$\texttt{E1M2}: \quad e = [p_2], \quad m = [p_1 p_0]. \tag{37}$$

We adopt a common, IEEE-like convention with bias $= 1$ for E2M1 and $= 0$ for E1M2 and subnormals when $e = 0$. Special values (Inf/NaN) are omitted, consistent with many FP4 deployments.

**Integer-like codebook mapping.** Under the above convention, E1M2 produces a nearly uniform magnitude set $\{0, 0.5, 1.0, \ldots, 3.5\}$ (ignoring sign). With a fixed scale $\alpha = 2$, this maps to

$$\alpha \cdot \{0, 0.5, 1.0, \ldots, 3.5\} = \{0, 1, 2, \ldots, 7\}, \tag{38}$$

and with sign yields $\{-7, \ldots, 7\}$, aligning with common symmetric INT4 quantization ranges. This provides an architectural bridge between float-like and integer-like FP4 semantics without changing the compute datapath.

## B.3. Zero-Overhead Metadata Packing: Type-in-Scale

A practical challenge is how to convey the FP4 sub-format (E2M1 vs. E1M2) without enlarging metadata. We exploit a simple observation: the NVFP4 scale is stored as an unsigned FP8 E4M3 value. Therefore its "sign" bit is either fixed to 0 or otherwise unused by the numerical encoding.

We repurpose the MSB of the 8-bit scale to carry a type bit:

$$\texttt{scale\_packed}[7] = \texttt{type},$$
$$\texttt{scale\_packed}[6{:}0] = \texttt{ue4m3\_mag}.$$

Hardware reconstructs the unsigned scale by forcing the sign bit to zero:

$$\texttt{scale\_ue4m3} = \{1'b0, \ \texttt{scale\_packed}[6{:}0]\}. \tag{39}$$

This achieves **zero additional storage and bandwidth cost** for type metadata. In our implementation, type is shared per-block to maximize amortization and simplify control.

## B.4. Overhead Analysis in NAND

We compare the incremental hardware cost of (i) enabling E1M2 in addition to E2M1 and (ii) implementing the block scaling machinery (scale-product + one per-block multiply). To make the analysis portable across synthesis flows, we express costs in NAND and provide parameterized formulas.

We model the total hardware overhead base on the estimation of previous work (Chen et al., 2025a).

$$n = min(2^{x+1} + 2y, \ psum\_bit\_width) \tag{40}$$

*Table 6.* Gate-complexity model for the MAC Unit with $k$ lanes. Here $x$ and $y$ denote exponent and mantissa widths; for INT, $x = 0$. The aligner width $n$ is given by Equation (40). "Main Cells" list dominant standard cells used in aggregation.

| Sub-block | INT Mul | FP Mul | INT Add | FP Add | Main Cells |
|---|---|---|---|---|---|
| Multiplier | $k(x+y+1)^2$ | $k(y+1)^2$ | – | – | AND, FA, HA |
| Adder (mantissa/int) | – | – | $2k(x+y+1)$ | $kn$ | FA, HA |
| Exponent adder | – | $kx$ | – | – | FA, HA |
| Exponent subtractor | – | – | – | $kx$ | XOR, FA, HA |
| Comparator | – | – | – | $kx$ | XOR, AND, OR |
| Aligner (barrel) | – | – | – | $kn \log_2 n$ | MUX |
| Normalizer (shared) | – | – | – | $n \log_2 n$ | MUX, OR |

### B.4.1. COST MODEL

We use standard architecture-level approximations:

$$G_{\text{NOT}} = 1 \, NAND, \tag{41}$$
$$G_{\text{AND2}} = 2 \, NAND, \tag{42}$$
$$G_{\text{OR2}} = 2 \, NAND, \tag{43}$$
$$G_{\text{HA}} = 5 \, NAND, \tag{44}$$
$$G_{\text{FA}} = 12 \, NAND, \tag{45}$$
$$G_{\text{MUX2}} = 2 \, AND + OR + NOT \tag{46}$$
$$= 7 \, NAND, \tag{47}$$

### B.4.2. INCREMENTAL COST: E1M2 SUPPORT

Relative to an E2M1-only decode (pure wiring of exponent/mantissa bits), dual-mode decode requires per FP4 element:

$$\Delta G_{\text{E1M2,elem}} \approx 2G_{\text{MUX2}} + 2G_{\text{AND2}} \approx 18 \, NAND. \tag{48}$$

The block-shared inversion $\neg t$ adds one inverter per block and is amortized by 8 lanes. For one block dot that needs decoders consumes both A and B operands (16 FP4 elements total), the incremental cost is:

$$\Delta G_{\text{E1M2,}block\text{,A+B}} \approx 16 \times 18 = 288 \, NAND, \tag{49}$$

plus a negligible constant number of inverters.

Convert E2M1 multiplier to E2M2 also increase the over head of adder between E2M2 multiplier, which increase the multiplier cost from $8 \times 4$ FAs to $8 \times 9$ FAs the adder cost from $8 \times 10$ to $8 \times 12$ FAs and aligner cost from $\approx 8 \times 30$ to $\approx 8 \times 40$ MUXs. (We assume MUX $\approx$ MUX2.)

Finally, the total estimate incremental cost of an accumulate vector is about:

$$\Delta G \approx 288 \ + \ 40 \, FA \ + \ 16 \, FA + 80 \, MUX = 288 \ + \ 480 \ + \ 192 \ + \ 560 = 1520 \, NAND \tag{50}$$

### B.4.3. LIMITATIONS

The hardware modeling is a coarse-grained estimate intended to capture only the order of magnitude of the predicted results. We do not model register overhead; while adding extra registers would reduce the reported overhead percentage of our method, it would also reduce the fidelity of the model.

### B.5. Hardware Design Details

### B.5.1. DECODER FOR MIXFP4

Figure 13 shows the decoder used for MixFP4. We repurpose the sign bit of the block scale as a block-shared type signal to select between an E2M1 shift path and an E1M2 lookup path, and both paths produce a unified E2M2 internal representation.

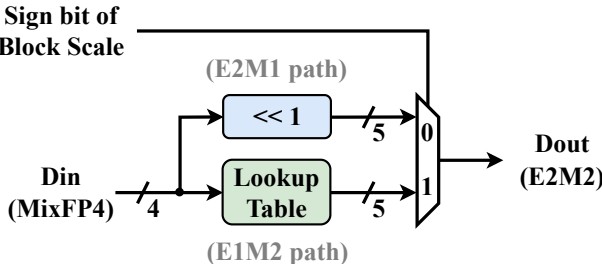

*Figure 13.* MixFP4 decoder. The block-scale sign bit selects between an `E2M1` shift path and an `E1M2` lookup path to produce a unified `E2M2` internal representation.

For `E2M1`, decoding is implemented as a simple bit extension (appending a trailing 0 bit), while `E1M2` uses a small fixed lookup table to map 4-bit payloads into the same internal format. This simple decoder unifies both inputs into a common `E2M2` internal representation, providing a uniform data format for the downstream compute datapath.

## C. Experimental settings of SmoothQuant, GPTQ and SpinQuant integration

### C.1. SmoothQuant

We utilize the training split of the `wikitext-raw-v1` dataset as the calibration set for SmoothQuant. Specifically, we calibrate each model using 512 samples, with each sample containing 512 tokens. The migration strength hyperparameter $\alpha$ is set to 0.5 for all related experiments.

### C.2. GPTQ

We adopt the GPTQ implementation from FP-Quant (Egiazarian et al., 2025). We select the training split of `wikitext-raw-v1` for calibration, using 1024 samples with a sequence length of 2048 tokens. Following the implementation in the FP-Quant repository, we enable activation quantization during the GPTQ process. Regarding the data format selection, we employ a *static* strategy: the optimal data format for each weight block is determined prior to the GPTQ algorithm based on the calibration data. Consequently, during the GPTQ error compensation phase, the selected formats remain fixed and are not updated by the error propagation process.

### C.3. SpinQuant

For SpinQuant, we also use the training split of `wikitext-raw-v1` for calibration. We adhere to the original settings described in the SpinQuant repository (Liu et al., 2024b), optimizing rotation matrices over 100 training steps, with each step consisting of 2048 tokens.

For the post-rotation GPTQ step, we use the original configuration from the SpinQuant repository, which differs from the setting described in Appendix C.2. Specifically, activations are *not* quantized during this process. Furthermore, the format selection strategy is *dynamic*: the error updating process quantizes weights and selects the data format that yields the minimal Mean Squared Error (MSE) on the fly.

## D. Ablation Study of Stochastic Rounding

Stochastic rounding probabilistically rounds a value to one of its two nearest representable numbers, where the probability is inversely proportional to the distance. For an integer format (e.g., with a range of $[-7, 7]$), this process can be formulated as follows:

$$\tilde{q}_{\text{scaled}} = q_{\text{scaled}} + \epsilon - 0.5, \quad \text{where } \epsilon \sim \mathcal{U}(0, 1) \tag{51}$$

$$q = \text{clamp}(\text{round}(\tilde{q}_{\text{scaled}}), -7, 7) \tag{52}$$

Table 7 presents the ablation results on validation loss, measured over the final 2.45B to 2.5B seen tokens.

| Seen Tokens (B) | 2.454 | 2.459 | 2.464 | 2.469 | 2.475 | 2.480 | 2.485 | 2.490 | 2.495 |
|---|---|---|---|---|---|---|---|---|---|
| BF16 | 3.7376 | 3.6300 | 3.6222 | 3.6696 | 3.6539 | 3.6427 | 3.5881 | 3.6514 | 3.6795 |
| NVFP4 | 3.8051 | 3.6964 | 3.6873 | 3.7319 | 3.7183 | 3.7068 | 3.6518 | 3.7163 | 3.7458 |
| + SR | 3.8055 | 3.6947 | 3.6868 | 3.7313 | 3.7190 | 3.7077 | 3.6523 | 3.7182 | 3.7446 |
| 4/6 | 3.7941 | 3.6850 | 3.6770 | 3.7222 | 3.7069 | 3.6986 | 3.6436 | 3.7072 | 3.7357 |
| + SR | 3.7933 | 3.6849 | **3.6747** | 3.7224 | 3.7076 | 3.6986 | 3.6427 | 3.7063 | 3.7349 |
| MixFP4 | 3.7942 | 3.6866 | 3.6787 | 3.7215 | 3.7071 | 3.6962 | 3.6409 | 3.7070 | 3.7339 |
| + SR | **3.7919** | **3.6839** | 3.6748 | **3.7193** | **3.7047** | **3.6940** | **3.6381** | **3.7047** | **3.7320** |

*Table 7.* Ablation study on the effectiveness of stochastic rounding.

# E. Block-wise format selection visualization

We present the Block-wise format selection visual result with a candidate set of {FP4 `E2M1`(6), FP4 `E1M2`, FP4 `E3M0`}.

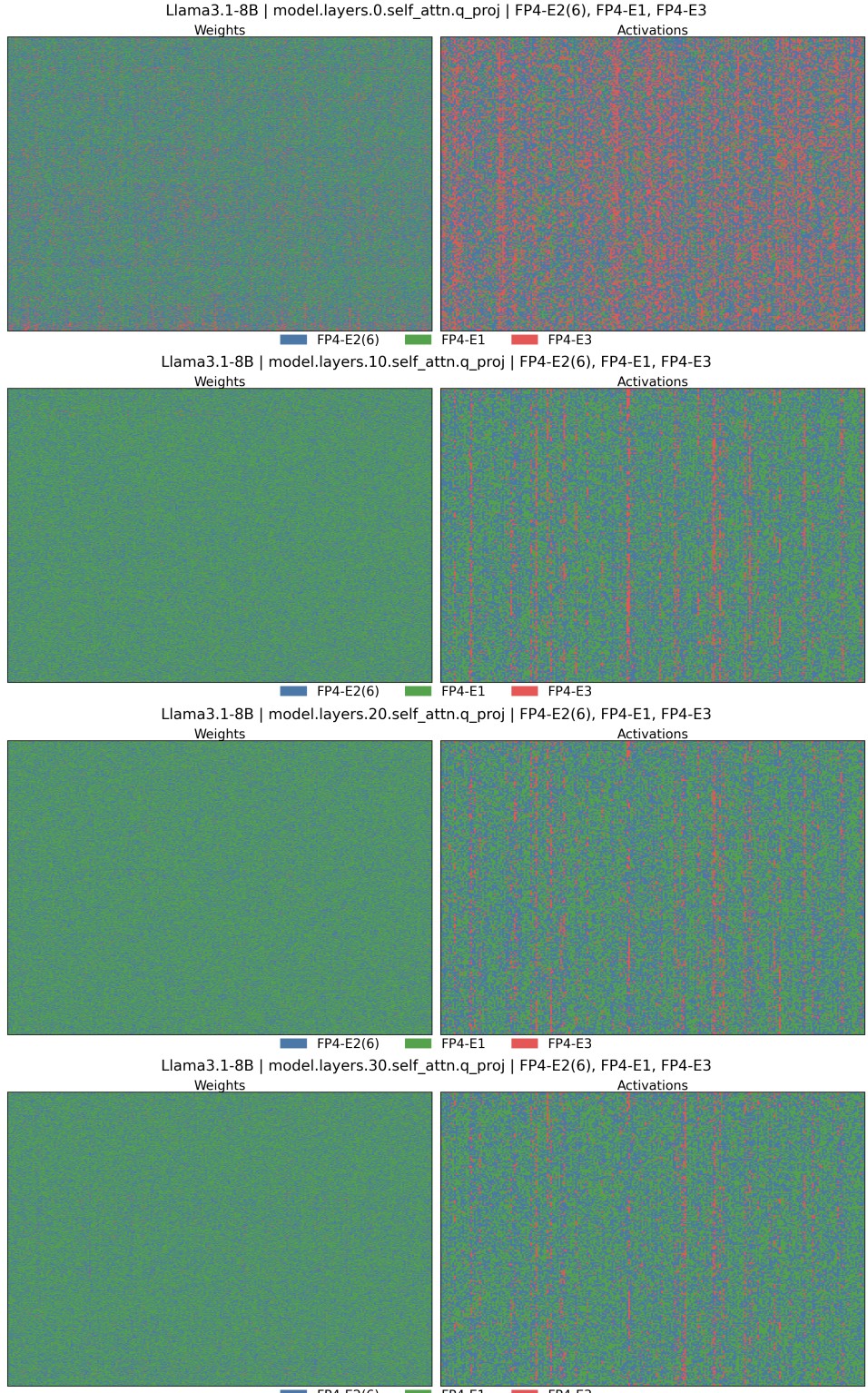

*Figure 14.* Block-wise format selection in Llama3.1-8B's Q projection modules of layer 0, 10, 20, 30.

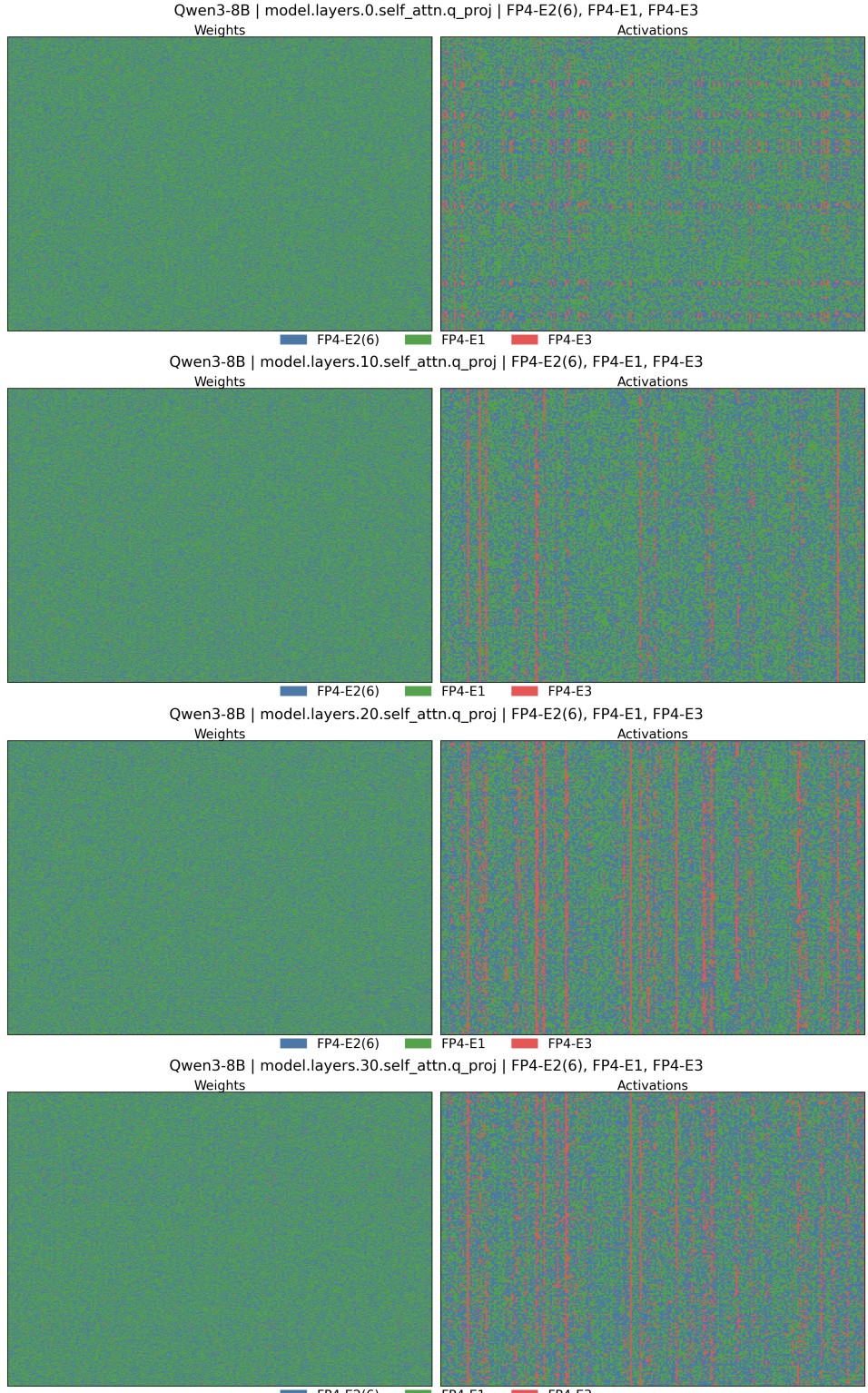

*Figure 15.* Block-wise format selection in Qwen3-8B's Q projection modules of layer 0, 10, 20, 30.

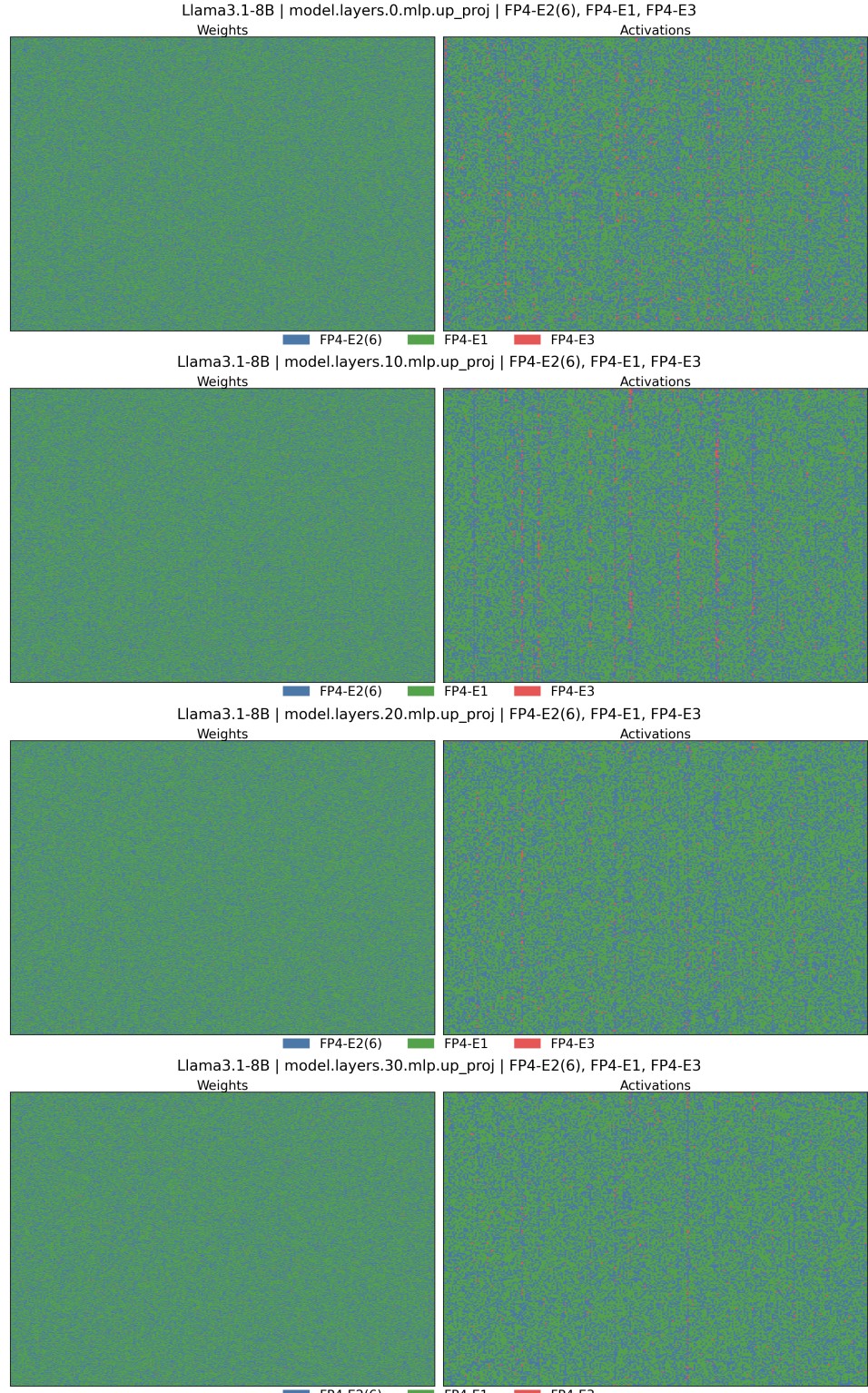

*Figure 16.* Block-wise format selection in Llama3.1-8B's UP projection modules of layer 0, 10, 20, 30.

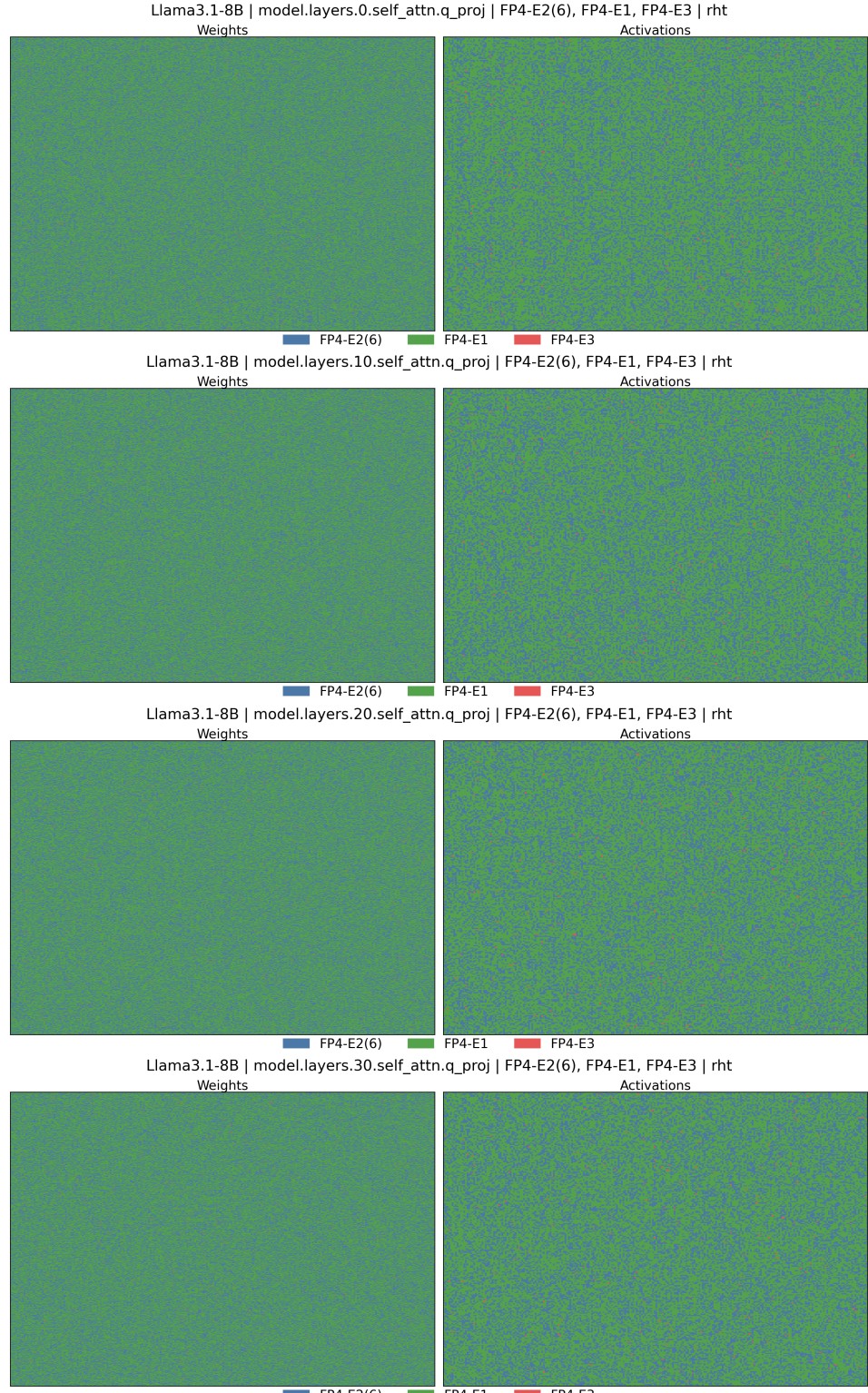

*Figure 17.* Block-wise format selection in Llama3.1-8B's Q projection modules of layer 0, 10, 20, 30 with RHT.

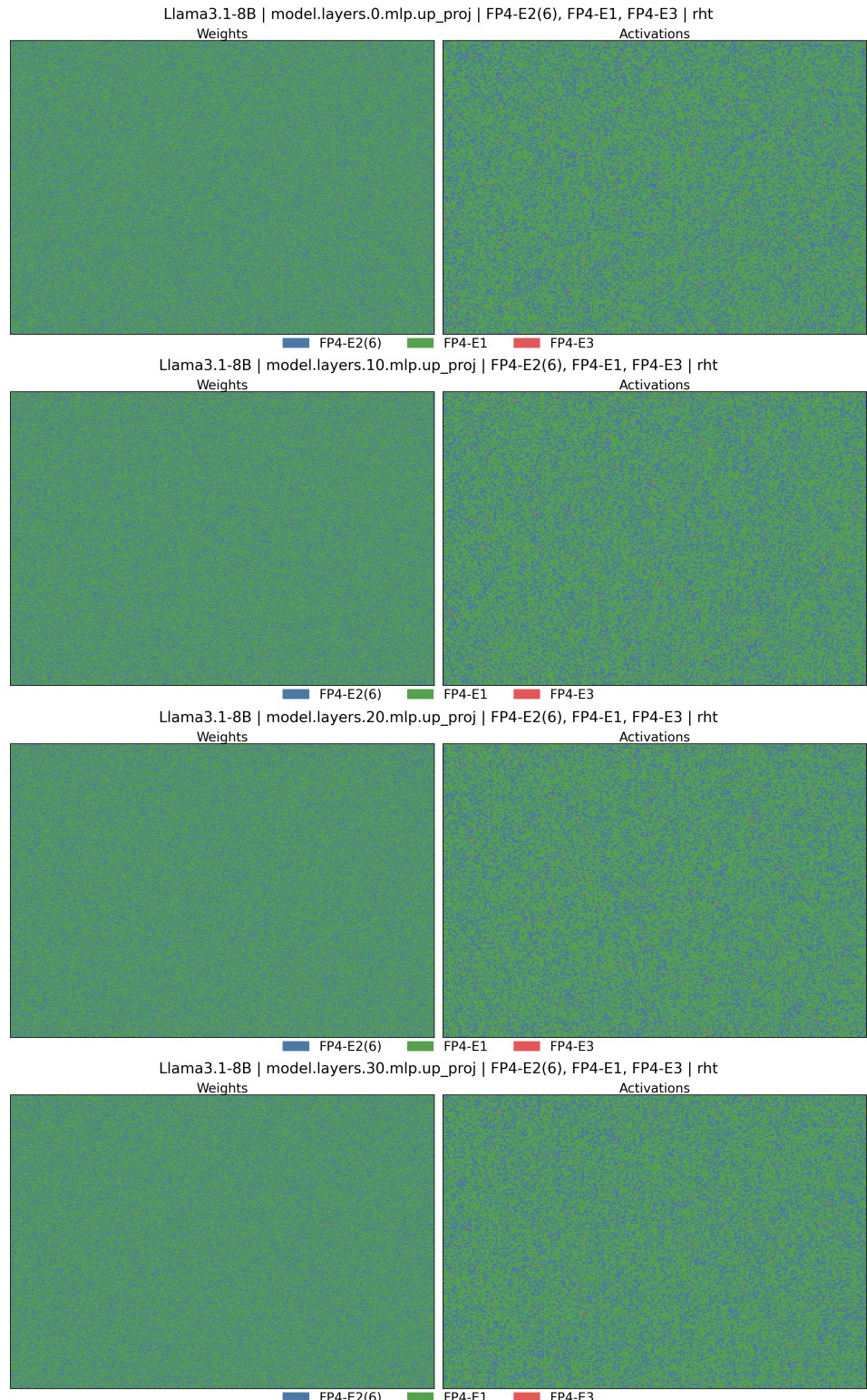

*Figure 18.* Block-wise format selection in Llama3.1-8B's UP projection modules of layer 0, 10, 20, 30 with RHT.

