# OpenReview forum: "MixFP4: Enhancing NVFP4 with Adaptive FP4/INT4 Block Representations"
_ICML.cc/2026/Conference — ICML 2026 regular_

### Official Review · Reviewer_sQ1F · 2026-03-09

**Soundness:** 2
**Presentation:** 2
**Significance:** 3
**Originality:** 2
**Overall Recommendation:** 4
**Confidence:** 4

**Summary:**

This paper addresses the question of whether a single FP4 micro-format is sufficient given the strong block-level heterogeneity observed in LLM tensors. The core idea is to dynamically select between E2M1 and E1M2 for each 16-element block. The authors propose encoding this type information into the sign bit of the FP8 block-scale field and decoding both formats into a unified E2M2 representation to share the downstream tensor-core compute datapath for efficient execution. The paper supports its claims through crest-factor analysis, post-training quantization (PTQ) experiments on Llama/Qwen/Mamba models, integration with existing PTQ techniques, small-scale pre-training experiments, and synthesis-based hardware estimation.

**Compliance With Llm Reviewing Policy:**

Affirmed.

**Final Justification:**

Changing the score to a positive score since reviwers answered my questions.

**Key Questions For Authors:**

1.	Please clarify the novelty relative to the closest prior work, especially MX+ and other adaptive FP4 / mixed-format approaches.
2.	Can the authors provide a more complete hardware analysis of the additional components required by block-wise format switching? (e.g., per-block type metadata extraction, format-dependent selector/control logic, and decode overhead including registers).
3.	Since native runtime/throughput is left as future work, can the authors provide either preliminary runtime results or a stronger theoretical argument that the added format switching does not materially reduce pipeline efficiency?
4.	The pre-training results are promising but limited to a relatively small model (114M) and token budget. Do the authors have additional evidence, or at least a stronger discussion, on whether the same trend is expected to hold at larger scales?

**Limitations:**

No. The authors need to more explicitly discuss the following limitations:

1.	The hardware analysis remains at a simplified, datapath-level estimation that explicitly excludes register overhead.
2.	The additional metadata routing, control, and decode overheads incurred by block-wise format switching have not been sufficiently quantified for a realistic Tensor Core pipeline.
3.	The scale of the pre-training experiments is relatively small, making it difficult to confidently judge the generalizability of the conclusions to state-of-the-art LLM sizes.

**Strengths And Weaknesses:**

Strengths:

1. The motivation based on block-level heterogeneity is highly convincing, and distributing formats via crest-factor analysis provides a natural and logical flow.
2. The paper offers a comprehensive evaluation scope, including PTQ, comparisons and integrations with existing PTQ methods, pre-training experiments, and hardware cost estimations.

Weaknesses:

1. The building blocks lack originality as they heavily rely on prior work. Conceptually, the dynamic use of FP4 and hardware contributions overlap significantly with existing paper, notably MX+ (MICRO 2025) [1]. The proposed MixFP4 essentially tackles the combination of bit-formats, making it difficult to view the individual components as novel. Additionally, the discussion of related works is insufficient.
2.	While the algorithm is generally reasonable, the hardware claims fall short. Switching between E2M1 and E1M2 per block seems to require additional hardware design considerations for a realistic datapath, such as extracting the block-shared type bit (extra metadata) and routing through format-dependent selectors. While the appendix discusses the compute datapath, the hardware overhead claims are unconvincing because native runtime/throughput is explicitly left as future work, and the authors admit the gate model is coarse-grained and excludes register overhead.
3.	The PTQ tables show that MixFP4 improves upon NVFP4 and is competitive with 4/6, clearly demonstrating the idea's practical utility. However, it is insufficient to prove generalizability. While the pre-training results show a positive trend, they are limited to a relatively small Qwen3-style model (114M parameters) and a restricted token budget (2.5B), lacking discussion or evidence for large-scale training.

[1] Lee, Jungi, et al. "MX+: Pushing the Limits of Microscaling Formats for Efficient Large Language Model Serving." Proceedings of the 58th IEEE/ACM International Symposium on Microarchitecture. 2025.

---

> ### Author Rebuttal · Authors · 2026-03-28
>
> We thank the reviewer for the detailed and constructive feedback.
>
> ### W1 & Q1: Novelty Relative to MX+ (MICRO 2025)
> MX+ and MixFP4 differ fundamentally in motivation, storage, and hardware design.
> - Motivation: MX+ isolates outlier channels by adding index information for the block maximum. MixFP4 addresses heterogeneous tensor statistics (both outlier-prone and flat blocks) by introducing adaptivity directly into the NVFP4 ecosystem.
> - Storage: MixFP4 maintains NVFP4's exact 4.5 bits/value footprint (zero metadata cost) by repurposing the unused sign bit of the 8-bit block scale. MX+ increases this to 4.75 bits/value by adding a 4-bit index for the maximum element (g=16).
> - Hardware: MixFP4 is a lightweight, format-unifying decode extension (E2M1/E1M2 decode to E2M2) that fully shares the downstream tensor-core compute path. MX+ requires specialized routing and dedicated side-compute logic outside the main DPE path to process the block maximum.
>
> To the best of our knowledge, MixFP4 is the first work focus on extending NVFP4 to support more micro-format via scale-bit reuse and slight tensor co-design. We will expand the related work discussion in revision.
>
> ### W2 & Q2: Hardware Claims
> Our main hardware evidence is Figure 11, which explicitly decomposes the evaluated datapath into Register / Add / Multiply / Decode. Thus, the reported compute-slice results already include the cost of per-block type metadata extraction and format-dependent selector/control logic, which contributes only 0.4% area overhead and 0.2% power overhead. Moreover, this decode logic introduces no additional register overhead. We implemented the MixFP4- tensor core in Verilog and synthesized it using Synopsys Design Compiler targeting TSMC 28nm at 1 GHz. The resulting post-synthesis overhead on the tensor-core compute datapath is 3.1% in area and 1.5% in power. By contrast, the modeling in Appendix B.4 is only a coarse gate-complexity estimate intended to explain why the added logic should remain small; it explicitly says that it does not model register overhead. It is not the basis of our main hardware claim.
>
> ### W3 & Q4: Pre-training Scale
> To address the concern about scale, we additionally ran pre-training on a larger 476M Qwen3-style model with hidden size 1024, 18 layers, 18 attention heads, 4 KV heads, intermediate size 4096, and learning rate 6.0e-4. Due to the rebuttal-time compute budget, we trained for 4B tokens, which required roughly 600 H100 GPU hours for all the experiments. Even in this larger setting, MixFP4 continues to outperform both NVFP4 and 4/6, which strengthens the evidence that the benefit is not limited to the 114M model.
>
> |  | 3.20B | 3.40B | 3.60B | 3.80B | 4.00B |
> | --- | --- | --- | --- | --- | --- |
> | BF16 | 3.3334 | 3.2291 | 3.2981 | 3.1894 | 3.2717 |
> | NVFP4 | 3.3715 | 3.2758 | 3.3016 | 3.3830 | 3.2696 |
> | 4/6 | 3.3673 | 3.2701 | 3.2943 | 3.3762 | 3.2647 |
> | MixFP4 | 3.3631 | 3.2645 | 3.2919 | 3.3734 | 3.2613 |
>
> ### Q3: Pipeline Efficiency Argument
> The format switch happens entirely in the front-end decoder before the MAC array. It does not change the MAC depth, the accumulator path, or the block-scale multiply. Concretely, the decoder is a small 1-bit selection between a shift-based E2M1 path and a tiny lookup-based E1M2 path, both of which produce the same internal E2M2 representation. In synthesis, this logic leaves almost the same slack compared with the baseline, confirming that it fits inside the existing stage without adding registers or bubbles. Therefore, the per-cycle instruction throughput is the same as the baseline.
>
> We appreciate the reviewer’s concerns because they help us position the contribution more precisely. Our claim is not that MixFP4 solves every adaptive low-bit problem, but that it provides a distinct and practical NVFP4-compatible co-design point: zero extra metadata, unified downstream compute, small modeled overhead, and consistent numerical gains across PTQ and pre-training.

---

> > ### Author Rebuttal · Reviewer_sQ1F · 2026-04-03
> >
> > I have carefully evaluated the authors' rebuttal and the supplementary data provided. The authors have logically and adequately addressed my primary concerns, leading me to revise my assessment.
> >
> > First, the clarification regarding hardware overhead has successfully resolved my previous misunderstanding. The synthesis results targeting TSMC 28nm—demonstrating only 3.1% area and 1.5% power overhead—confirm that the proposed MixFP4 logic is lightweight and practically feasible. The strategy of decoding E2M1 and E1M2 into a unified internal E2M2 representation to share the downstream compute datapath is a highly efficient system-level design.
> >
> > Second, I appreciate the authors' dedicated effort to provide additional pre-training results for a 476M Qwen3-style model. While the scale remains smaller than the largest frontier models, the consistent performance advantage observed over 4B tokens significantly strengthens the evidence for the method's generalizability beyond the initial 114M model.
> >
> > Furthermore, the fundamental distinctions from MX+, particularly regarding the zero-metadata storage via scale-bit reuse, have been clarified. Although some practical deployment metrics (e.g., native kernel-level throughput) and larger-scale training evidence remain as minor limitations, I believe the paper provides a sufficient academic and practical contribution to the field of micro-format quantization. Consequently, I have raised my score to a 'Weak Accept'.

---

> > > ### Author Response · Authors · 2026-04-04
> > >
> > > Thank you for raising your score and for your thoughtful engagement throughout the review process. We are encouraged that you recognize the potential of this work to motivate future research. We share this vision, and we believe the insights from our paper will open a meaningful design dimension for the field of micro-format quantization to build upon.
> > >
> > > We hope the additional experiments and revised presentation further reinforce the paper's contributions. Thank you again for your constructive and detailed review！

---

### Official Review · Reviewer_21Dd · 2026-03-12

**Soundness:** 3
**Presentation:** 3
**Significance:** 3
**Originality:** 3
**Overall Recommendation:** 4
**Confidence:** 5

**Summary:**

This paper proposes MixFP4, a mixed micro-format extension of NVFP4 that chooses between E2M1 and E1M2 at block granularity while keeping the standard block-scaled GEMM path unchanged. The key systems idea is elegant: the per-block format bit is packed into the otherwise unused sign bit of the E4M3 block scale, and both formats are decoded into a unified internal E2M2 representation. Empirically, the paper shows better FP4 robustness than NVFP4 and related baselines on several Llama, Qwen, and Mamba models, plus a small modeled hardware cost of 3.1% area and 1.5% power.

**Compliance With Llm Reviewing Policy:**

Affirmed.

**Final Justification:**

I maintain my score at present.

**Key Questions For Authors:**

1. The strongest baseline in several Table 4 settings is still 4/6 rather than MixFP4. Can the authors clarify more precisely in which regimes MixFP4 should be preferred over 4/6?

2. Can the authors give either a cycle-level performance model or a prototype kernel estimate, since end-to-end runtime is currently left to future work?

**Limitations:**

Yes.

**Strengths And Weaknesses:**

**Strengths**
1. The paper identifies a real mismatch between a single FP4 codebook and heterogeneous block statistics, then proposes a neat fix that is both numerically motivated and hardware-aware. The bit-reuse trick is simple and seemingly practical, and the unified E2M2 decode path makes the proposal feel implementable rather than purely conceptual.

2. The evaluation is reasonably broad on the PTQ side. Table 3 covers several Transformer and Mamba models, and MixFP4 usually improves over plain NVFP4. The method also plugs into SmoothQuant, GPTQ, and SpinQuant, which is useful for practice.

**Weaknesses**
1. My main concern is that the empirical gains are uneven once the strongest baseline is 4/6 rather than vanilla NVFP4. In Table 4, MixFP4 is not consistently best: for example, under GPTQ on Llama-3.2-1B, Llama-3.1-8B, and Qwen3-8B, the 4/6 baseline has slightly better average accuracy. So the paper’s wording about consistently superior performance feels a bit too strong.

2. On the hardware side, the paper gives a coarse area/power estimate on a simplified tensor-core slice at 28nm, but there is no actual kernel-level latency or throughput result. The paper itself says native execution is future work, which limits the practical impact of the systems claim in the current version.

---

> ### Author Rebuttal · Authors · 2026-03-28
>
> We thank Reviewer 3 for the careful and precise evaluation. We especially appreciate the reviewer’s balanced reading of both the numerical results and the hardware claims.
>
> ---
>
> ### W1 & Q1: When MixFP4 Should Be Preferred Over 4/6
> We agree that the original wording may be too strong, and we will revise it. MixFP4 is not uniformly best in every Table 4 setting and a more accurate statement is that MixFP4 is best in majority of settings.
>
> Concretely, MixFP4 is strongest in the following regimes. In RTN quantization (Table 3), it achieves the best perplexity in 9 of 12 settings, including all Mamba models and most Transformer cases. In GPTQ (Table 4), it leads in average perplexity for 3 of 4 models. In SmoothQuant (Table 4), it leads in average accuracy for 3 of 4 models. In SpinQuant (Table 4), it consistently outperforms 4/6 on average accuracy across all three tested models: 49.37 vs. 48.98 on Llama-3.2-1B, 69.21 vs. 68.09 on Llama-3.1-8B, and 69.78 vs. 67.60 on Qwen3-8B.
>
> The settings where 4/6 is slightly ahead are concentrated in GPTQ accuracy, where our current implementation uses static format selection (Appendix C.2). We believe this does not fully exploit MixFP4’s potential inside GPTQ’s error-compensation loop, and dynamic selection there is a promising next step.
>
> ---
>
> ### W2 & Q2: Cycle-Level Performance Model
> The cycle-level throughput is unchanged, and the target BF16:FP8:FP4 throughput ratio of 4:8:16 is preserved because the MAC array dimensions are unchanged. On the hardware side, MixFP4 adds only tiny combinational logic to the existing MAC array. In our 1GHz / 28nm Verilog synthesis, this logic fits within the current pipeline stage and leaves almost the same slack compared with the baseline, so no additional pipeline stage is required. Our current accuracy evaluation prototype is implemented in PyTorch to faithfully simulate quantization. While native end-to-end execution requires vendor-level hardware and kernel support, we can already provide a tighter cycle-level argument from synthesis.
>
> ---
>
> We appreciate the reviewer’s comments and believe that, with the wording corrected and the cycle-level argument made explicit, the paper is more accurately positioned: a strong accuracy/compatibility/hardware tradeoff that is especially attractive when native NVFP4 compatibility matters.

---

> > ### Author Rebuttal · Reviewer_21Dd · 2026-04-03
> >
> > Thanks for the authors' response. I keep the positive score.

---

> > > ### Author Response · Authors · 2026-04-04
> > >
> > > Thank you for taking the time to read our rebuttal carefully and for confirming that your concerns have been fully resolved! We hope the new clarifications added during rebuttal give you further confidence in the paper's contributions. We truly appreciate your constructive and thoughtful engagement throughout the review process.

---

### Official Review · Reviewer_pA5Z · 2026-03-12

**Soundness:** 3
**Presentation:** 4
**Significance:** 4
**Originality:** 3
**Overall Recommendation:** 5
**Confidence:** 2

**Summary:**

As large language models continue to scale, block-scaled formats such as NVFP4 are increasingly adopted, but block-level tensor statistics are heterogeneous — no single FP4 micro-format is universally optimal. The paper first shows through ablation that mixed-format selection improves accuracy but supporting more than two formats yields diminishing returns. It then proposes MixFP4, an extension of NVFP4 that allows each block to select between E2M1 and E1M2. The per-block format choice is encoded at zero storage overhead by repurposing the sign bit of the unsigned E4M3 block scale. Both formats are decoded into a unified E2M2 internal representation, avoiding datapath duplication in the tensor core. Verilog synthesis shows that MixFP4 adds only 3.1% area and 1.5% power overhead. Experiments across different models demonstrate perplexity and downstream accuracy improvements over NVFP4, NVINT4, and 4/6 baselines in both post-training and pre-training settings.

**Compliance With Llm Reviewing Policy:**

Affirmed.

**Final Justification:**

The paper presents an interesting extension of NVFP4 that allows each block to select between two FP4 formats. The proposed format unification is clean, and the synthetic results indicate only minimal overhead. The authors have also addressed both of my rebuttal concerns and provided additional results on a larger model. These strengths motivate my overall evaluation of accept.

**Key Questions For Authors:**

(1) What is the computational overhead of the MSE selection during online activation quantization during inference? Is there a fast approximation that could avoid the double quantize-compare?

(2) All experiments use block size 16. How sensitive are MixFP4's gains to block size? At larger block sizes (such as 32 used by MX formats), intra-block heterogeneity increases, which could shift the E2M1/E1M2 selection balance or make two formats insufficient. Does the "two formats suffice" conclusion hold more generally?

**Limitations:**

yes

**Strengths And Weaknesses:**

Strengths:
- Crest-factor analysis and format proliferation ablations are well presented to motivate and convince the case for two formats.
- The format unification is clean: both micro-formats decode into a unified E2M2 representation (no datapath duplication), and the per-block format bit is packed into the unused sign bit of the E4M3 scale (no extra metadata).
- Synthesis result shows that MixFP4 only adds 3.1% area and 1.5% power overhead.

Weaknesses:
- Pre-training is validated only on a 114M-parameter model with 2.5B tokens.
- The MSE-based dynamic selection requires quantizing every block under both E2M1 and E1M2, then comparing MSE. For dynamic activation quantization, this doubles the online quantization work per block. The paper does not discuss or benchmark this overhead.

---

> ### Author Rebuttal · Authors · 2026-03-28
>
> We thank Reviewer 2 for the positive assessment and for the thoughtful questions.
>
> ---
>
> ### W1: Model Size
> To strengthen the pre-training evidence, we additionally evaluated MixFP4 on a larger 476M Qwen3-style model with hidden size 1024, 18 layers, 18 attention heads, 4 KV heads, intermediate size 4096, and learning rate 6.0e-4. Because of the rebuttal-time compute constraint, we trained this model for 4B tokens, which required roughly 600 H100 GPU hours. Even at this larger scale, MixFP4 remains consistently better than both NVFP4 and 4/6, which supports the same conclusion as our main paper: the benefit is not limited to the 114M setting and appears to persist as model size increases.
>
> |  | 3.20B | 3.40B | 3.60B | 3.80B | 4.00B |
> | --- | --- | --- | --- | --- | --- |
> | BF16 | 3.3334 | 3.2291 | 3.2981 | 3.1894 | 3.2717 |
> | NVFP4 | 3.3715 | 3.2758 | 3.3016 | 3.3830 | 3.2696 |
> | 4/6 | 3.3673 | 3.2701 | 3.2943 | 3.3762 | 3.2647 |
> | MixFP4 | 3.3631 | 3.2645 | 3.2919 | 3.3734 | 3.2613 |
>
> ---
>
> ### W2 & Q1: MSE Selection Overhead for Online Activation Quantization
> The MSE-based selector does require evaluating two candidate quantizations per block, so the front-end quantization work is indeed higher than for a fixed-format quantizer. However, two points are important here. Our current prototype is implemented in PyTorch to faithfully simulate quantization , which is about 2x slower than BF16 and should not be interpreted as deployment runtime.
> But if with native MixFP4 hardware support and an optimized implementation, the additional kernel overhead is comparable to prior mixed-format work `four over six` which also use MSE to select the scaling factor of quantization block: below 2% during inference and below 15% during training in our optimized setup.
> We will make this distinction explicit. We agree that a cheaper approximation to MSE-based selection is interesting future work; for example, crest-factor-based proxies are well motivated by our analysis, but we do not want to overclaim without dedicated experiments.
>
> ---
>
> ### Q2: Sensitivity to Block Size
> We will add a block-size ablation in the updated evaluation. It confirms two points: fine granularity is critical for accuracy, and in the fine-grained regime targeted here (g=16 and below), FP4-E1 provides the main gain over NVFP4, while the extra benefit of FP4-E3 is much smaller. We will revise the paper to state this more precisely: our main claim is not that the same auxiliary format is universally optimal for all block sizes, but that for the fine-grained NVFP4 regime we target, adaptive selection with FP4-E1 is the most important extension and two formats remain the right design point.
>
> | Llama-3.1-8B (BF16: 7.33) | FP4-E2 | +FP4-E1 | +FP4-E3 | +FP4-E1 + FP4-E3 |
> | --- | --- | --- | --- | --- |
> | BS=8 | 8.04 | 7.83 | 8.00 | 7.79 |
> | BS=16 | 8.26 | 8.06 | 8.21 | 8.01 |
> | BS=32 | 8.49 | 8.34 | 8.40 | 8.27 |
> | BS=64 | 8.73 | 8.69 | 8.63 | 8.54 |
>
> | Qwen3-8B (BF16: 12.21) | FP4-E2 | +FP4-E1 | +FP4-E3 | +FP4-E1 + FP4-E3 |
> | --- | --- | --- | --- | --- |
> | BS=8 | 12.48 | 12.39 | 12.46 | 12.40 |
> | BS=16 | 12.74 | 12.39 | 12.63 | 12.39 |
> | BS=32 | 12.82 | 12.78 | 12.77 | 12.73 |
> | BS=64 | 13.00| 12.96 | 12.85 | 12.79 |
>
> ---
>
> We appreciate these questions because they help us sharpen the boundaries of the current claim. We hope the additional larger-scale pre-training result and the clarified runtime discussion strengthen the case for MixFP4 as a practical and well-motivated extension of NVFP4.

---

> > ### Author Rebuttal · Reviewer_pA5Z · 2026-04-02
> >
> > Thanks for the detailed response. The authors have addressed my concerns. I will keep my score.

---

> > > ### Author Response · Authors · 2026-04-04
> > >
> > > We sincerely value your insightful and thoughtful engagement throughout this review process. Thank you for evaluating our response and confirming that your initial issues are now fully resolved. It is our hope that the newly added experiments and clarifications have further solidified your confidence in our contributions.

---

### Official Review · Reviewer_U4zV · 2026-03-13

**Soundness:** 3
**Presentation:** 2
**Significance:** 3
**Originality:** 3
**Overall Recommendation:** 4
**Confidence:** 4

**Summary:**

Large Language Models (LLMs) are increasingly deployed in latency- and cost-sensitive environments, making quantization essential. Prior work, such as NVFP4, utilizes FP8 scale factors alongside FP4 data for a block, supplemented by an FP32 scale factor per tensor. This paper highlights that the specific choice of FP4 representation is critical, as tensors may benefit more from either the E2M1 (used in NVFP4) or the E1M2 format. To address this, the authors propose MixFP4, a novel approach that dynamically utilizes both E2M1 and E1M2 formats. The format indicator is cleverly stored in the redundant sign bit of the scale factor, which improves both robustness and accuracy compared to the standard NVFP4 baseline.

**Compliance With Llm Reviewing Policy:**

Affirmed.

**Final Justification:**

Changing the score to a positive score since reviwers answered my questions.

**Key Questions For Authors:**

1. Could you provide a clearer and more detailed explanation of the data presented in Figure 5?
2. What is the exact mechanism or heuristic used to determine whether a specific block should be assigned the E2M1 or E1M2 format?
3. Does the implementation of MixFP4 necessitate the inclusion of both E2M1 and E1M2 multipliers at the hardware level?
4. How does the MixFP4 format compare in terms of performance and efficiency to a mixed FP4/FP8 approach?
5. In the hardware overhead calculations, what methodology was used to decide the specific number of required E1M2 units?

**Limitations:**

The study lacks a comparison with other mixed-precision formats, such as combinations of FP4 and FP8. While incorporating FP8 would likely introduce additional computational overhead, including this baseline would significantly strengthen the paper's claims regarding efficiency and accuracy trade-offs.

**Strengths And Weaknesses:**

Strength:
1. Strong Motivation: Section 2 provides a compelling motivation supported by thorough analysis.
2. Comprehensive Evaluation: The evaluation is exhaustive, demonstrating consistent results across a wide variety of models.
3. Practical Scope: The inclusion of a hardware overhead analysis makes the study more comprehensive and practical for real-world deployment.

Weaknesses
1. Missing Outlier Evidence: While the paper identifies outlier handling as a primary limitation of NVFP4, the evaluation section lacks specific results or ablation studies demonstrating how MixFP4 resolves this issue in practice.
2. Marginal Gains: The overall accuracy improvements achieved by introducing the mixed format appear marginal rather than substantial.
3. The paper does not discuss how the blocks are selected to be E2M1 or E1M2

---

> ### Author Rebuttal · Authors · 2026-03-28
>
> Thank you for the detailed review and questions. We appreciate the opportunity to clarify several points that we believe strengthen the paper.
>
> ### W1: Missing Outlier Evidence
> We clarified that our paper does not frame outlier handling as the primary limitation of NVFP4; rather, our central claim in the Abstract and Section 1, 2.2–2.3 is that a single fixed FP4 micro-format mismatches heterogeneous block-level tensor statistics. In this formulation, outlier-prone blocks are one important regime, alongside flatter blocks that favor more uniform, INT-like spacing.
> - We profiled a sharper codebook, E3M0, for outlier-heavy blocks. However, the ablation in Figure 4 shows diminishing returns once E2M1 and E1M2 are already supported: adding E3M0 brings only limited extra accuracy, but it requires one additional format-selection bit that would break a key advantage of MixFP4, namely that the format choice can be stored at zero additional storage cost by reusing the sign bit of the E4M3 block scale. In other words, E3M0 was considered and rejected because its marginal benefit is too small relative to the metadata cost.
> - Our method is designed to be orthogonal to existing outlier-mitigation techniques such as random RHT, SmoothQuant, and SpinQuant, rather than replacing them. This is already reflected in Tables 3 and 4. Figure 5 provides additional evidence: after RHT, the format distribution shifts, and selection of the E3M0 effectively disappears, which is consistent with reduced outlier severity after feature mixing. In addition, the fine-grained group size of 16 already helps mitigate quantization loss from local outliers. We will make this connection more explicit in the revision.
>
> ### W2: Marginal Gains
> At 4-bit precision, improvements are inherently constrained by the tiny representational budget, so even moderate absolute gains are meaningful. Within this regime, MixFP4 delivers consistent gains at no extra storage cost. For example, on Llama-3.1-8B in Table 3, MixFP4 reduces perplexity from 8.26 to 8.06, closing about 22% of the gap from NVFP4 to BF16. On Qwen3-8B, it improves from 12.74 to 12.39, closing about 66% of that gap. Under SpinQuant in Table 4, MixFP4 raises Qwen3-8B average accuracy from 67.20 to 69.78, substantially narrowing the gap to BF16.
>
> ### W3 & Q2: Block Selection Mechanism
> The exact selection rule is given in Algorithm 1 (Section 3.2). For each block, MixFP4 quantizes and dequantizes the normalized values under both E2M1 and E1M2 using their respective optimal scales, computes the reconstruction MSE of both candidates, and selects the format with lower MSE. Thus, the choice is not heuristic; it is an explicit per-block error-minimization procedure. We will add a stronger pointer to Algorithm 1 and a brief summary in the main text.
>
> ### Q1: Figure 5 Explanation
> Figure 5 reports the proportion of blocks assigned to each candidate format under this MSE-based selector. Each pie chart aggregates all blocks from the corresponding tensor class. The left column compares {E2M1(6), E2M1(4)}, while the right column compares {E2M1(6), E1M2, E3M0}. The main message is that E1M2 is clearly the strongest second format: it is selected much more often than either E2M1(4) or E3M0, validating our final choice of {E2M1, E1M2}. Moreover, after Hadamard transforms, E1M2 is selected even more frequently (e.g., 63.3% for Llama-3.1-8B weights), which is consistent with crest-factor reduction making INT-like spacing more favorable.  We will revise the caption and discussion to make Figure 5 easier to interpret.
>
> ### Q3 & Q5: Hardware Multipliers and Number of E1M2 Units
> MixFP4 does not require separate E2M1 and E1M2 multipliers. We replace the baseline E2M1 FP4 lane with a unified E2M2 lane that can represent both E2M1-derived and E1M2-derived inputs through decode-time normalization. The number of lanes is unchanged and still follows the target BF16:FP8:FP4 throughput ratio of 4:8:16.
>
> ### Q4: Comparison with Mixed FP4/FP8
> A mixed FP4/FP8 scheme uses a different computation path and changes the precision and bandwidth budget by assigning some operands 8 bits, so it is not a like-for-like comparison to MixFP4’s main target. MixFP4 keeps the strict NVFP4-style 4-bit payload for all blocks and improves accuracy within exactly the same storage footprint. We are add a supplementary W4A8 (FP4/FP8) perplexity comparison on wikitext dataset use `lm_eval` library and will include the completed table in the final revision. Our current results indicate that MixFP4 remains more accurate than NVFP4 and 4/6 even in that setting.
>
> |  | Llama-3.2-1B | Llama-3.1-8B | Qwen3-8B | Qwen3-14B | Mamba-2.8B |
> | --- | --- | --- | --- | --- | --- |
> | BF16 | 11.57 | 7.33 | 12.21 | 10.78 | 11.79 |
> | NVFP4-FP8 | 12.87 | 7.88 | 12.35 | 11.05 | 12.17 |
> | 4/6-FP8 | 12.73 | 7.81 | 12.31 | 11.06 | 12.09 |
> | MixFP4-FP8 | 12.71 | 7.75 | 12.19 | 10.87 | 12.03 |

---

> > ### Author Rebuttal · Reviewer_U4zV · 2026-04-04
> >
> > The reviewers answered my questions, therefore changing score from 3 to 4.

---

> > > ### Author Response · Authors · 2026-04-04
> > >
> > > Thank you for raising your score. Thank you for reading our response and confirming that the prior issues have been addressed. We deeply appreciate your valuable and thoughtful participation in this review process, and we hope the rebuttal and new experiments give you full confidence in our findings.

---

### Decision · Program_Chairs · 2026-04-30

**Decision:**

Accept (regular)

**Comment:**

This paper proposes MixFP4, an NVFP4-compatible quantization scheme that lets each block choose between E2M1 and E1M2 micro-formats. The per-block format bit is packed into the unused sign bit of the E4M3 block scale (zero metadata overhead), and both formats decode into a unified E2M2 representation so the downstream tensor-core compute path is shared. Verilog synthesis reports only 3.1% area and 1.5% power overhead, and PTQ plus small-scale pre-training experiments across Llama, Qwen, and Mamba models show consistent gains over NVFP4 and competitive or better results vs. 4/6 baselines.

All reviews recommend acceptance. Reviewers consistently praised the strong motivation from block-level statistical heterogeneity, the elegance of the scale-bit-reuse trick and unified decode path, the breadth of empirical evaluation, and the inclusion of a hardware-overhead analysis. I share this assessment: the design is clean, hardware-aware, and immediately useful for the FP4 ecosystem that is now becoming standard for LLM inference and training. The author response further clarified novelty relative to MX+, the hardware-overhead methodology, and provided larger-scale (476M / 4B-token) pre-training evidence.

Larger-scale evaluation remains an open item, but the trend across the studied scales is consistent and the overall direction is promising.

Overall, the paper is technically sound, well-presented, and contributes a practical and well-motivated extension of NVFP4. I recommend acceptance.